# DNA affects the phenotype of fuel-dependent coacervate droplets

Corbin Machatzke[1,3], Anna-Lena Holtmannspötter[2,3], Hannes Mutschler ®[1] ✉ & Job Boekhoven ®[2] ✉

Synthetic cells emulate fundamental biological behaviors, like growth, metabolism, and mobility, but have lacked genotype-driven selection, which is essential for Darwinian evolution. Here, we introduce libraries of short DNA sequences as genotypes into fuel-dependent peptide-RNA-based coacervate droplets, serving as synthetic cells. By sequencing, we identify sequences that partition in the droplets, revealing strong preferences for guanine-rich and adenine-rich motifs. These sequences affect the synthetic cell phenotype—adenine-rich sequences shorten droplet lifetimes through hybridization. In contrast, guanine-rich sequences kinetically trap droplets via peptide interactions, altering dissolution rates and morphology. This study demonstrates how genotype affects phenotype in synthetic cells, establishing essential design principles for achieving Darwinian evolution in minimal protocellular systems.

Synthetic cells are minimal models of living systems[1,2], designed from the bottom up to replicate fundamental biological behaviors such as growth[3–7], division[8–13], metabolism[14–16], and evolution[17]. A hallmark of life is the ability to sustain itself out of equilibrium through continuous energy transduction, replicate and adapt under selective pressures, ultimately enabling Darwinian evolution[18–23]. Thus, for a synthetic cell to ultimately become synthetic life, it must sustain under non-equilibrium conditions and demonstrate Darwinian evolution, *i.e.*, replication with mutation, such that the fittest survive and spread their genotype. To address the criterion of synthetic life that requires self-sustaining properties, we recently introduced the concept of fuel-dependent synthetic cells[24–26]. These synthetic cells are driven by a chemical reaction cycle in which a fuel-driven activation reaction activates building blocks, which then spontaneously deactivate through hydrolysis[27–31]. Thus, the synthetic cells can grow under conditions where chemical fuel is abundant but decay and dissolve during periods of starvation (Fig. 1a). These fuel-dependent synthetic cells exhibit life-like properties as they require continuous energy input to maintain their structure and functionality[24]. Without fuel, these synthetic cells spontaneously decay, serving as a selection pressure, especially in experiments in which fuel is periodically supplied[12].

In previous work, we used fuel-dependent synthetic cells based on lipids[26] and complex coacervate droplets[24]. Complex coacervate droplets have emerged as promising protocell models due to their ability to spontaneously form membraneless compartments, concentrating biomolecules, and facilitating biochemical reactions[32–36]. Our fuel-dependent complex coacervates for the basis of this work comprise a long RNA polyanion (poly-U, 2200 monomer units on average) and a short cationic peptide (Ac-F(RG)₃D-OH, Fig. 1b). The fuel-dependent nature of these droplets comes from this peptide's fuel-driven activation. Specifically, the peptide's C-terminus reacts with carbodiimides as fuel, forming the corresponding intramolecular anhydride and increasing the overall charge of the peptide from +1 to +3. In the deactivation reaction, the short-lived anhydride spontaneously hydrolyzes to the original peptide with a half-life of roughly 1 minute, thereby reverting the charges from +3 to +1 (Fig. 1b). The transient increase in charges enables the short-lived activated peptide to bind the RNA and assist in forming complex coacervate-based droplets. Thus, upon addition of a batch of fuel, the chemical reaction cycle initiates, leading to a population of transiently activated peptides that form droplets via phase separation. Activated peptide diffuses into these droplets, leading to growth, while deactivated peptide leaves,

¹TU Dortmund University, Dortmund, Germany. ²Department of Bioscience, School of Natural Sciences, Technical University of Munich, Garching, Germany. ³These authors contributed equally: Corbin Machatzke, Anna-Lena Holtmannspötter. ✉e-mail: Hannes.Mutschler@tu-dortmund.de; Job.Boekhoven@tum.de

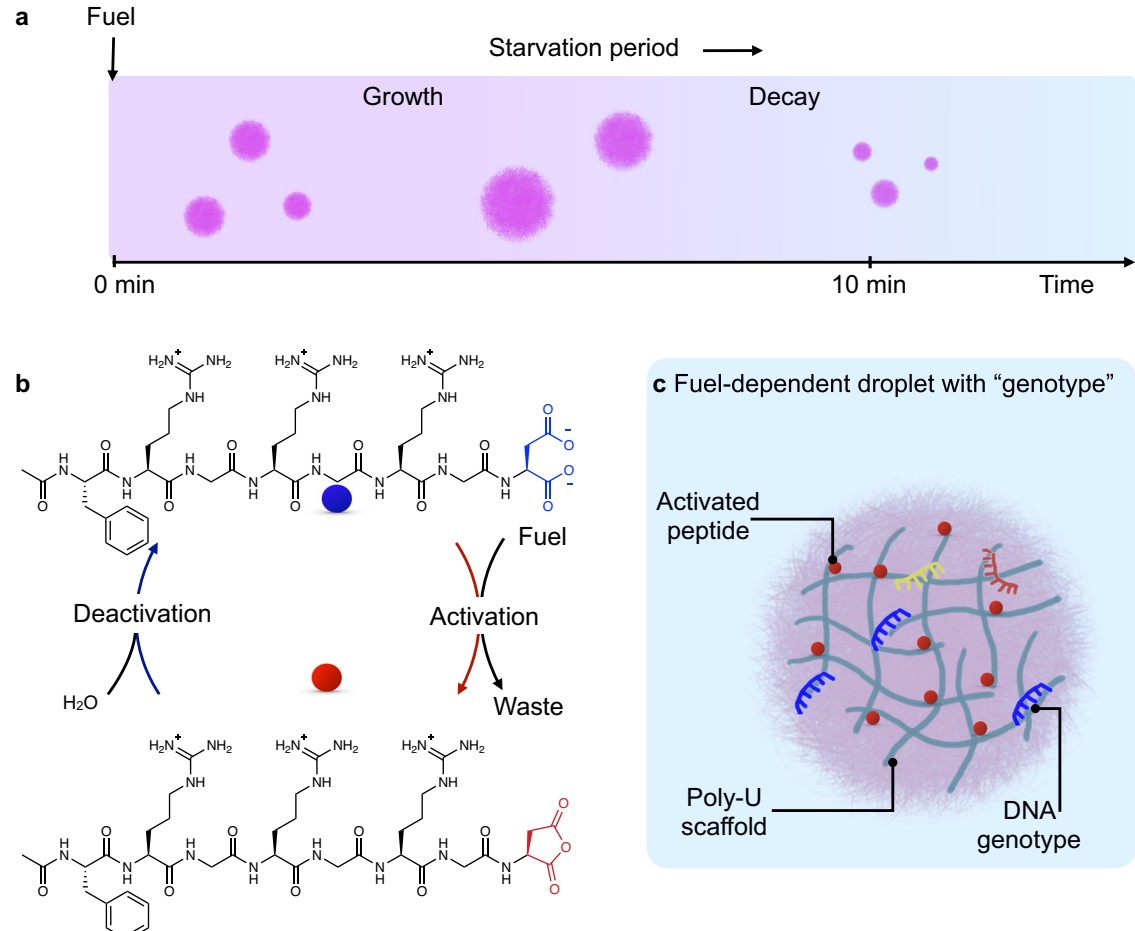

**Fig. 1 | Fuel-dependent synthetic cells. a** Upon fuel addition, droplets form, grow, fuse, and dissolve after a discrete lifetime. **b** Fuel-dependent peptide activation and deactivation cycle. Blue stands for the inactivated peptide with the aspartic acid C-terminus. Red stands for the activated peptide with a terminal anhydride. **c** Synthetic cells are equipped with a short DNA genotype (colored strands) that can interact with both the peptide (red) and the RNA scaffold material (gray).

leading to shrinkage. This battle for survival continues until no more fuel remains and all droplets dissolve (Supplementary Movie 1). So, like life, our fuel-dependent synthetic cells require a constant influx of nutrients, and periodic fueling of our droplets can sustain them.

A critical limitation of these droplets is the absence of information-carrying elements—a genotype—which means survival is purely stochastic. In synthetic life, fuel-dependent synthetic cells would require an equivalent of a genotype: inherited molecular information that can be replicated and passed on during division. This genotype should influence the system's physical or functional traits (its phenotype), thereby enabling selection and evolution. We use these biological terms with caution—in our synthetic context, "genotype" refers to a set of molecules that are inherited, replicable, and transferable to the next generation. At the same time, "phenotype" denotes the observable physical traits of the synthetic cell. Genotype–phenotype coupling, in this context, means that these information-containing molecules directly influence the properties of the droplets. This would be a much more rudimentary analog to the biology's genotype-phenotype coupling, which operates through the central dogma of molecular biology. Despite its simplicity, it could still enable principles like natural selection or even evolution, which is what we aim for in future work.

In this work, we propose that a set of single-stranded DNA sequences may serve as a proxy for such a genotype. We reason that DNA sequences can be replicated either with or without enzymes[17].

Besides, as the droplet material comprises RNA and partly peptide, we hypothesize that the DNA sequences could interact either with the RNA or peptide and thereby affect the droplet properties, i.e., their phenotype (Fig. 1c)[37]. Thus, we endowed our synthetic cells with short, 30-base DNA strands, starting from a library of random sequences, and investigated their compatibility and interaction modes with the droplets. After we found that libraries of 30-mers biased towards one of the four nucleotides can affect droplets, we used second-generation Illumina sequencing to identify which DNA sequences preferentially partition into fuel-dependent droplets. We found patterns in the sequencing data, which we then isolated and investigated individually. Specific sequences, such as long repeats of adenosine or guanosine, can drastically alter the droplet properties, increasing their longevity, viscosity, and decreasing fusion rates; in other words, a genotype affects the droplet's phenotype. However, the genotype has not yet replicated, and in future work, we envision using self-replicating DNA sequences as a genotype that also affects phenotype[38–40].

## Results

To establish a rudimentary genotype-phenotype coupling, we first investigated the impact of DNA libraries on the droplet's phenotype focusing on their lifetime and morphology (Fig. 2a). Throughout this study, we used the following standard reaction conditions unless stated differently: 23 mM peptide, 4.1 mM poly-U (concentration expressed per nucleotide), 30 mM EDC, and 50 μM DNA (30-mer

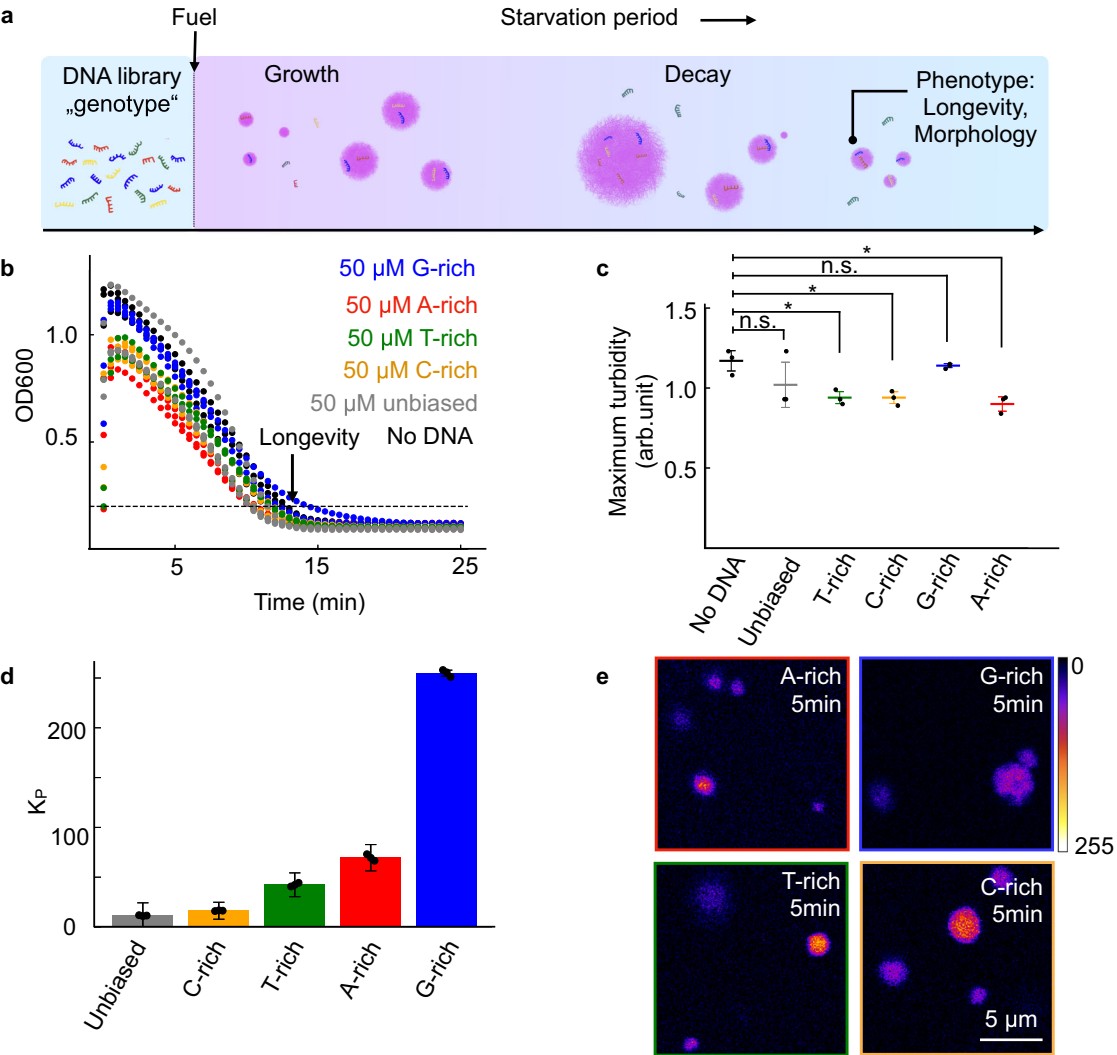

**Fig. 2 | Biased DNA libraries as genotype impact the droplet phenotype.**
**a** Schematic representation of the droplet's lifecycle. **b** Absorbance of 600 nm light (OD 600) against time as a measure of turbidity for standard conditions in the presence of 50 μM of different DNA libraries, color indicates a bias towards one nucleobase: black meaning no DNA, gray no bias, blue a bias for guanine, red adenine, green thymine, and yellow cytosine. The same allocation applies to the other subfigures. **c** Comparison of the maximum turbidity (arbitrary unit) for different DNA libraries. The A-rich library limits coacervate formation the most, while the G-rich library has no impact. **d** Comparison of the partitioning coefficients ($K_P$) of biased random sequences into our droplets. The G-rich random sequences and A-rich random sequences have the highest partitioning. **e** Representative Confocal

micrographs with 500 nM Sulforhodamine. Scale bars represent 5 μm. Error bars in Fig. 1c show the standard deviation from the average ($N = 3$). $N$ represents one biological replicate of a turbidity value. Significance was determined with a two-sided Welch's t-test with the no-DNA sample as the control group. $P$-values are available in Supplementary Table 1. Error bars in Fig. 1d are calculated from the standard deviation from the average ($N = 3$) of three biological replicates of measured DNA concentration in the supernatant (shown datapoints) and the standard deviation from the average of the droplet volumes calculated from confocal images of water in oil droplets ($N = 10$) according to the method described in the method section. Source data are provided as a Source Data file.

oligonucleotides, corresponding to a total DNA concentration of 1.5 mM when expressed per nucleotide). We used DNA libraries consisting of random 30-mer sequences, with each nucleotide position having an equal (25%) probability of containing any of the four canonical bases (A, T, C, or G). Additionally, we used biased DNA libraries, in which one specific nucleotide (A, C, T, or G) was preferentially incorporated at each position of the 30-mer sequence with a probability of 76%, while the remaining three bases each had a 8% chance of incorporation.

In a plate reader, we tracked the OD600 of the samples over time as a measure of the presence of the droplet population (Fig. 2b). We found that most samples with DNA tended to exhibit less turbidity compared to samples without DNA, except for the unbiased and G-rich library, which showed no significant difference (Fig. 2c, Supplementary Table 1). This finding suggests fewer or smaller droplets due to the

added DNA. We defined the time at which the OD600 fell below 0.2 as the longevity of the droplet population and found that the A-rich library showed a trend toward shorter longevity (Fig. S1, Supplementary Table 2). Taken together, among all the libraries, the A-rich library proved the most destructive for the droplets, resulting in less turbidity that was shorter-lived. We further investigated other droplet properties, such as viscosity and resistance to salt in the presence of the unbiased library, which remained unchanged (Figs. S2, S3). Lastly, we examined the impact of the biased DNA libraries on droplet morphology by confocal microscopy (Fig. 2e, Fig. S4). The droplets appeared round, rapidly fused, and remained unchanged overall, regardless of the DNA.

For DNA to act as a genotype, it must incorporate into the synthetic cells. Thus, we tested the partitioning of the biased DNA libraries into the droplets. As the partitioning strongly depends on whether a

droplet is dissolving or growing (Fig. S5), these experiments with our short-lived droplets are challenging. Therefore, we used an in-equilibrium model of our droplets to determine the partitioning coefficients. Specifically, a stable model of the activated peptide was used (Ac-F(RG)$_3$N-NH$_2$). In the model of the activated peptide, the C-terminal aspartic acid is substituted with an amidated asparagine, resulting in three positive charges, similar to the activated peptide, but without its transient nature. We replaced 3 mM of the peptide with the active peptide model to form droplets at a time point roughly midway through the cycle[41]. We determined the partition coefficient by measuring the DNA concentration in the supernatant using HPLC after spinning down the droplets (Supplementary Table 7) and by calculating the droplet volume using confocal microscopy. We found that completely random DNA libraries (N$_{30}$) partitioned weakly (K$_P$ = 11) (Fig. 2d). We suspect that partitioning in this case is somewhat low due to competition with poly-U, which competes for a limited amount of anhydride. The short DNA is disadvantaged compared to the longer poly-U due to multivalency effects. The C- or T-biased DNA libraries did not partition substantially better than the unbiased DNA. In contrast, the partition coefficients increased by more than 20-fold for G-rich libraries and by more than 6-fold for A-rich libraries. We assume that partitioning for both is driven by interactions with the droplet material. While the A-rich DNA could base pair with our RNA scaffold, the G-rich DNA could interact strongly with our highly concentrated peptide. From our combined experiments, we learned that G-rich and A-rich DNA sequences have the highest partitioning coefficient. The A-rich library decreased the amount of droplet material formed and slightly decreased their longevity. In contrast, the G-rich library is the only one that does not limit droplet formation compared to the no-DNA condition.

We suspected that the minute changes in the droplet's phenotype are a result of the DNA libraries only partitioning weakly in the droplets. To put that in perspective, for the unbiased library, only 5% of the DNA sequences can be expected to be present in the droplets based on the partitioning coefficient (K$_P$ = 11) and the total droplet volume. That number increases to roughly 54% percent for the G-rich library. In other words, only a fraction of our DNA library acts as a genotype, in that it partitions in the droplets and affects their properties. To understand which DNA sequences those are, and unveil patterns in these sequences, we examined the sequences of the unbiased DNA library present in the spun-down, combined droplets and those in the supernatant using Illumina sequencing (Fig. 3a). Specifically, we prepared active droplets in the presence of the unbiased DNA library at a concentration of 0.7 μM 30-mer. After adding fuel, we waited for one minute to enable maximum droplet formation, followed by three minutes of centrifugation to separate the droplets from the supernatant. We recovered DNA from both the pellet and the supernatant and prepared the sequencing library using a custom protocol. That library was sequenced using Illumina sequencing (see Method section for detailed methodology). All subsequent analysis was performed in a comparative approach between the R1 reads of each sample, ensuring that no bias in the library preparation method leads to false results.

The sequencing results revealed no significant difference between the supernatant and the original library, further strengthening our hypothesis that only a small fraction of the library partitions in the droplets (Fig. S6). In contrast, we observed drastic differences between the sequences in the combined droplets and the supernatant. For example, we found an enrichment of the G nucleotide—of all the nucleotides in the droplets, there was a 2.5% G-enrichment compared to the supernatant (26.5% in the supernatant, 29 % in the combined droplets, Fig. 3b). This finding is in line with our finding that G-rich libraries partitioned strongly compared to unbiased 30-mer libraries. Excitingly, we could now unveil which nucleotide positions in the 30-mer these G predominantly occupied. We found a positional preference for guanines in the terminal bases: both 5' and 3' ends

contained Gs with an increased likelihood of 8% points compared to the supernatant (33% against 25%). The enrichment diminishes over the successive eight nucleotides, reaching a plateau of ~1.5% (Fig. 3c). When we study the sequences with sequential Gs at their termini, we found they have a bias for more Gs in the rest of the sequence compared to all sequences from the droplet sample, e.g., a sequence starting with 3 Gs in a row, tends to have more Gs in the remaining 27 bases compared to the rest of the pool (Fig. S7). The latter finding suggests a cooperative effect—for a sequence to partition, it is more effective to have multiple Gs within a single sequence.

We calculated and compared the minimum free energy of folding for all sequences in either the combined droplets or the supernatant using a thermodynamic secondary structure prediction algorithm[42–44]. The droplets exhibit a shift towards less folded sequences, characterized by a shifted distribution in the minimum free energy of folding. (Fig. 3d). Based on the thermodynamic secondary structure prediction, we conclude that intramolecular structures in sequences are detrimental to their partitioning into these droplets in line with previous work[45,46].

Finally, we searched for enriched motifs in the droplet library compared to the supernatant using STREME[47–49]. We found 14 motifs that were statistically significantly enriched compared to the supernatant (Supplementary Table 9). Of these sequences, we present the four sequences with the highest significance in the main text as frequency logos (Fig. 3e). Two of these four are dominated by multiple Gs in consecutive stretches having a high probability of five or even nine consecutive Gs. The other two of the four logos contain stretches of consecutive As, at least five nucleotides long, with a slight trend of this motif towards the 3' or 5' ends of the sequences (Supplementary Table 9). We further analyzed the frequency of stretches of five and seven consecutive As or Gs in both the droplets and the supernatant and found a substantially higher occurrence of those in the droplets compared to the supernatant (Fig. 3f).

Based on these findings, we conclude that multiple consecutive Gs are required to interact with the droplet material. Similarly, and in line with previous work[46], consecutive A sequences strongly interact with the poly-U in the droplets. Hybridization as a recruiting mechanism and stretches of consecutive bases have previously been found in the sequencing of RNA in different coacervate systems[50].

From the data above, we conclude that sequences with stretches of A or G partition well in the droplets. Moreover, these sequences likely also affect the droplet phenotype, as observed with the slight effects of A- and G-rich biased libraries. To test which sequences specifically affect the phenotype, we ordered 30-mers containing stretches of A and measured their partitioning and impact on droplet phenotype. We started with A$_{30}$ as an extreme case of consecutive As (Fig. 4a). Compared to the unbiased or the A-rich library, A$_{30}$ resulted in a drastic decrease in droplet lifetime (Fig. 4b, Supplementary Table 3, Supplementary movie 2), which was strongly concentration-dependent above a concentration as low as 20 μM (Fig. 4c and Fig. S8). We found that the partition coefficient was nearly five hundred-fold higher for A$_{30}$ compared to the A-rich random sequences, implying that over 99% of all A$_{30}$ oligomers are in the droplets (Fig. S9). The high partition coefficient was to be expected[51], as A$_{30}$ can hybridize with poly-U, but it does not explain the decrease in longevity.

We hypothesize that the hybridization of the short A$_{30}$ with the RNA causes the domains of the poly-U to rigidify, given that these domains are chimeric double helices (Fig. 4e)[52]. As the flexibility of the RNA is essential for efficient charge screening and phase separation[53–56], this could explain the change in the droplet's phenotype. We examined the critical fuel concentration required for coacervate formation (Fig. 4d) and the critical salt concentration for droplet dissolution (Fig. S10). This fuel concentration corresponds to the anhydride concentration in solution required for phase separation. Without additional DNA, our coacervate-based droplets formed at

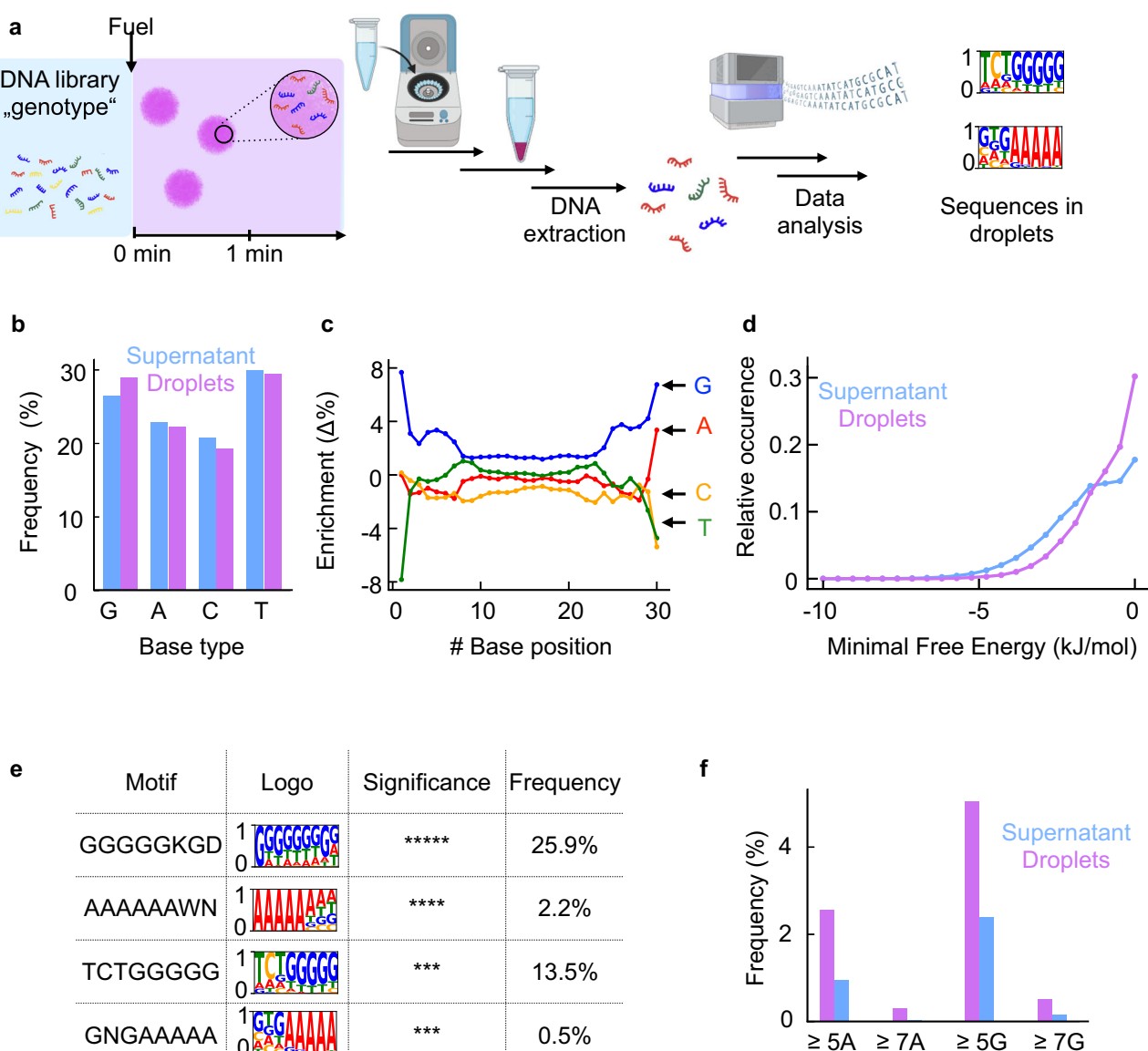

**Fig. 3 | Sequencing the combined droplets genotype. a** Schematic representation of our workflow: DNA was harvested from the combined droplets phase and the supernatant, then sequenced and analyzed. Schematic of the workflow created with https://BioRender.com/yf9mg24. **b** Overall base distribution in the supernatant (blue) and combined droplets phases (pink). **c** Difference in nucleotide content per position between sequences isolated from combined droplets and supernatant. Blue stands for guanine, red for adenine, green for thymine, and yellow for cytosine. The same allocation applies to subfigure **e**. **d** Minimal free energy of folding for sequences in the supernatant (blue) and combined droplets (pink) calculated with ViennaRNA package 2.0. **e** The four most overrepresented motifs in the combined droplets sample (pink) compared to the supernatant (blue). Stretches of at least 5 Gs or As seem to be highly favored. The reported significance are $e$-values which are $p$-values for each motif appearing at random in the dataset divided by the total number of motifs. Symbols signify $e$-values where ***** are values < $10^{-100}$; **** are values < $10^{-50}$; *** are values < $10^{-10}$. Exact $e$-values can be found in Supplementary Table 9. Significance was calculated with a Fisher's Exact test by MEME-Suites STREME as detailed in the software documentation. **f** The frequency of at least five or seven consecutive As or Gs in a sequence in the supernatant (blue) and in the combined droplet phase (pink). Lines in c and d are added to guide the eye. Source data are available on a public repository linked in the Data availability section.

9 mM EDC. In the presence of 50 µM $A_{30}$, the threshold increased to 27 mM, indicating a lower affinity between the droplet components. To determine the critical salt concentration, we used our in-equilibrium model droplets and monitored the amount of additional NaCl required to dissolve our droplets. Without DNA or 50 µM $N_{30}$, the critical salt concentration was 74 mM NaCl, which dropped to 54 mM NaCl in the presence of 50 µM $A_{30}$. This lower salt threshold further indicates that the additional $A_{30}$ reduced the interaction strength. We confirmed that poly-U is the primary interaction partner of $A_{30}$ by measuring the diffusivity of both the peptide and poly-U within droplets using FRAP. The poly-U mobility was markedly reduced, while the peptide diffusivity

remained unaffected with the additional DNA as genotype (Figs. 4f, g, and S11, Supplementary Table 11), indicating a selective restriction of the RNA. These results align with previous studies on peptides with poly-A/poly-U mixed systems[57], hybridization-induced diffusivity restriction[58], and other work, which shows the polymer chain stiffness is a limiting factor on the binodal in complex coacervation[56,59,60] and weakens the associative phase behavior of the system and its salt resistance[60]. Additionally, we have confirmed the poly-U adenine hybridization via native gel electrophoresis (Fig. S12) and investigated the impact of $A_{30}$ on droplets containing a different, non-hybridizing polyanion. Without hybridization, the lifetime was not impacted

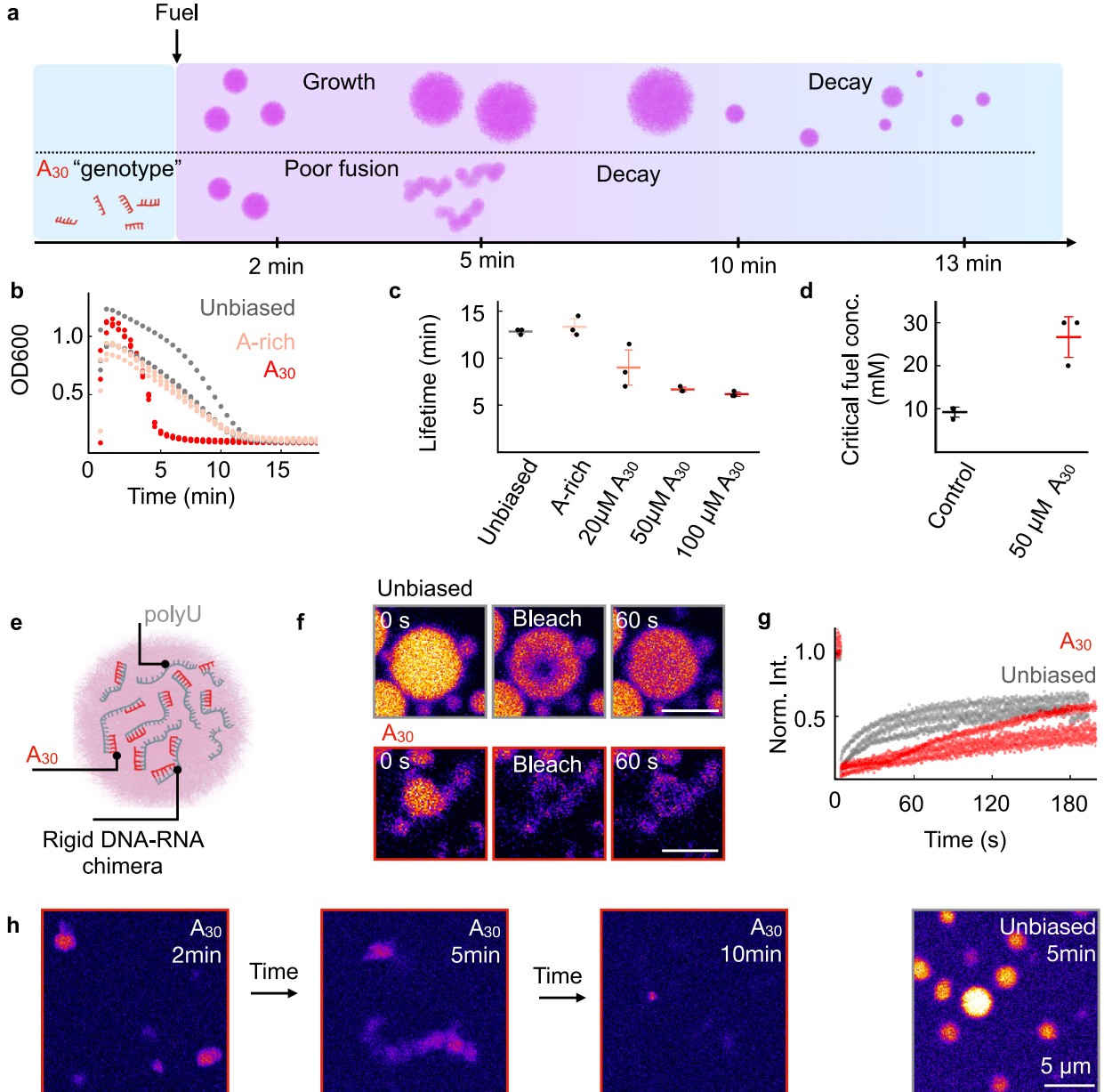

**Fig. 4 | Droplet's longevity decreases with a hybridizing genotype. a** Schematic representation of $A_{30}$ DNA strands (red) shortening the coacervate lifetime and changing the morphology. **b** Optical density at 600 nm (OD600) against time for droplets with an unbiased library (gray), an A-rich library (pink), or $A_{30}$ (red). **c** The lifetime of droplets with genotypes of unbiased DNA library (gray), A-rich library (pink), and different concentrations of $A_{30}$ (red, dark red). **d** The critical fuel concentration measured by tracking the OD600 upon fuel addition with (red) and without $A_{30}$ (black). **e** Schematic representation of hybridization of A30 (red strands) with the droplet material (gray). Representative confocal micrographs and FRAP recovery traces with normalized intensity (Norm. Int.) measured on a confocal microscope, bleaching 500 nM of Cy5-$A_{15}$ annealed to the poly-U in the presence of the unbiased library (gray) or $A_{30}$ (red). Scalebars represent $10\,\mu m$. **f, g** Confocal micrograph of select points in a fuel cycle stained with 500 nM Sulforhodamine in the presence of $A_{30}$ and $N_{30}$, scalebars represent $5\,\mu m$. **h** All error bars show the standard deviation from the average ($N = 3$). $N$ represents one biological replicate of one analyzed turbidity trace or turbidity value upon fuel addition. Source data are provided as a Source Data file.

(Fig. S13). Adding the A30 at a later point also shortened the lifetime, but did not lead to immediate dissolution (Fig. S14). Confocal microscopy over the droplet's lifetime showed that $A_{30}$-containing droplets fused less readily compared to their $N_{30}$ equivalents (Fig. 4h, Supplementary Movie. 2). Instead, they formed bead-like aggregates that are less liquid compared to our $N_{30}$ equivalents and resemble "pearls on a string" that settled onto the glass where they dissolved shortly after five minutes. We explain these results by the increased poly-U rigidity limiting the poly-U mobility and thereby preventing droplets from coalescing fully and growing. Nevertheless, the mobility of the peptide facilitates the dissolution of the aggregates when fuel levels are low. Similar morphologies have been reported in poly-A/poly-U mixtures and RNA-protein networks, based on base pairing, where the peptide or protein remains mobile despite the restricted dynamics of the RNA[61].

We conclude that DNA $A_{30}$ can hybridize with the RNA droplet material, stiffening the RNA and leading to radical changes in the droplet phenotype by rigidifying the RNA scaffold. The stark

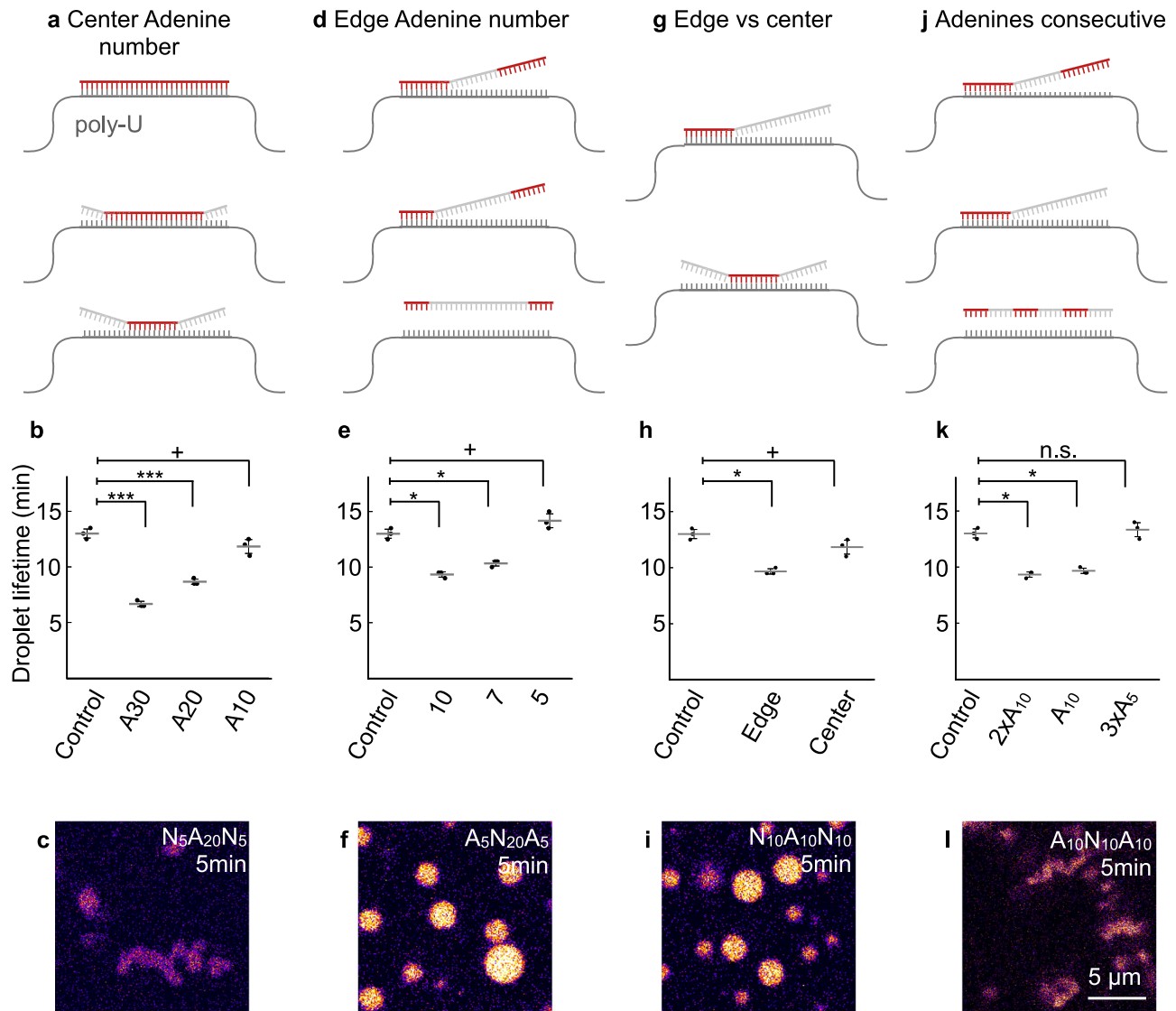

**Fig. 5 | Molecular design rules for genotype-phenotype relations for adenine.** Schemes of molecular design of different genotypes and their interaction with poly-U, red sections represent adenine stretches on an otherwise unbiased library strand (gray) (**a**, **d**, **g**, **j**). The lifetimes of droplets without a genotype and in the presence of specific sequences, determined by the OD 600 against time (**b**, **e**, **h**, **k**). All error bars show the standard deviation from the average ($N = 3$). $N$ represents one biological replicate of one analyzed turbidity trace. Significance was determined via a two-sided Welch's $t$-test with the no-DNA sample as the control group. $P$-values are available in Supplementary Table 4. Confocal micrograph in the presence of selected genotypes imaged 5 min after fuel addition on the glass using 0.5 mM NBD labeled peptide for staining (**c**, **f**, **i**, **l**). Source data are provided as a Source Data file.

difference in the droplet phenotype between $A_{30}$ and the A-rich DNA library implies that the genotype-phenotype coupling is sensitive. Not all A-rich sequences change the phenotype. Guided by our sequencing results, we set out to test the molecular design rules that dictate how DNA sequences affect the genotype.

Guided by the sequencing data, we investigated the effects of the position and length of an adenine stretch within the sequence. First, we tested the impact of homopolymeric adenine stretches on the droplet phenotype by comparing sequences containing stretches of ten, twenty, or thirty consecutive adenines positioned centrally within otherwise random oligomers (Fig. 5a). The introduction of ten adenines resulted in a marginal reduction of the droplet's lifetime. In comparison, twenty and thirty adenines led to a more pronounced decrease (Fig. 5b). This data suggests that longer adenine stretches enhance hybridization with poly-U, thereby decreasing droplet stability and changing morphology (Figs. 5c, S15). Interestingly, a central stretch of 20 adenines had a more substantial effect than a sequence

with 76% adenine content distributed randomly, despite both having comparable total A content. This finding suggests that not only the base composition but also the local sequence arrangement matters, *i.e.*, consecutiveness is more disruptive than random distribution.

Next, we designed DNA strands with adenine stretches located at both termini (Fig. 5d). We found that a total of ten adenines (five at each end) had no significant effect on the droplet lifetime or morphology, despite the presence of five consecutive adenines being a motif observed in the sequencing (Fig. 5e, f). Increasing the total number of terminal adenines to fourteen (seven per end) reduced the lifetime to 10.3 min, while twenty terminal adenines (ten per end) further shortened it to 9.3 min and also decreased fusion rates (Figs. 5e, S16). These results reinforce the observation that, irrespective of positioning, longer adenine stretches enhance interaction with poly-U and destabilize the droplets. In contrast, shorter stretches below a critical threshold of seven are insufficient to induce any phenotypical changes.

Having established that adenine stretches promote hybridization when positioned either centrally or at the termini of the sequence, we directly compared the impact of placing ten adenines at the ends versus in the middle (Fig. 5g). Terminal placement resulted in a more pronounced reduction in droplet lifetime—9.7 minutes—compared to 12.2 min for the central configuration (Figs. 5h, i, S17).

To assess the importance of uninterrupted consecutiveness, we compared sequences containing multiple stretches of consecutive As (Fig. 5j). This comparison revealed no significant difference in lifetime or morphology when adding a second stretch of ten consecutive adenines unconnected to the first (Figs. 5k, l, S18, Supplementary Table 4). We conclude that increasing the number of adenines while interrupting the consecutive stretch does not exacerbate the adverse effect further. The second stretch of adenines is most likely unable to bind to the poly-U strand due to the central sequence mismatch, and there is no crosslinking between the strands. We observe the same lack of impact when examining a sequence with fifteen adenines distributed in three blocks of five. We cannot circumvent the threshold of seven consecutive As by distributing stretches of five consecutive As. This suggests that phenotype modulation requires not merely interaction, but sufficiently tight and continuous binding to affect the droplet material.

Taken together, to induce phenotypic changes, at least 7 consecutive adenines are required, preferably at the edge of the sequence. This requirement likely accounts for the minimal lifetime shortening of the A-rich library. With this knowledge, we calculated, using a numerical sequence generation method, that roughly one-third of sequences in the A-rich library have seven or more consecutive adenines located within one base of either end. Introducing an alternative condition of twenty consecutive adenines anywhere in the sequence, we find that only 1.4% of sequences are compliant. This analysis suggests that only about one-third of the sequences in the biased random population could hybridize with the droplet scaffold, accounting for the weak effect on droplet properties. Repeating the calculation for the entirely random $N_{30}$ library yielded just 0.02% of sequences fulfilling either condition, in line with the low observed partitioning and the absence of any measurable influence on droplet behavior.

To assess whether specific G-rich sequences affect the phenotype, we investigated the effects of a sequence composed of eight guanines at both termini flanking a randomized central region of fourteen bases ($G_8N_{14}G_8$). Additionally, we included a $G_{10}A_{20}$ sequence in our analysis to examine the combined effects of guanine and adenine stretches on droplet behavior. The twenty adenines are expected to enable hybridization with poly-U, while the guanine stretch may introduce additional interactions.

Excitingly, the OD600 data of the active droplets with a genotype showed evidence of a favorable phenotype i.e., with 50 μM of $G_8N_{14}G_8$, the droplets exhibited increased longevity. In fact, the droplets no longer dissolved (Fig. 6a, b, Supplementary Table 5). We hypothesized that $G_8N_{14}G_8$ irreversibly sequestered the peptide within the droplets, as guanine exhibits a high affinity for arginine and adopts higher-order structures, such as G-quadruplexes, potentially enabling network formation[62,63]. Guanine-arginine contacts are abundant in RNA- and DNA-binding proteins due to pairwise hydrogen bonding as well as favorable cation-pi stacking[64–67]. In contrast, droplets with $G_{10}A_{20}$ dissolved showed a similar longevity as the unbiased library. The absence of kinetic trapping in $G_{10}A_{20}$ may result from the opposing effects of adenine-induced droplet destabilization counteracting guanine-mediated retention. Isothermal titration calorimetry showed a substantial difference in the dissociation constant of the peptide for both $G_8N_{14}G_8$ and $G_{10}A_{20}$ compared to the unbiased library and $A_{30}$, verifying the interaction between the G-rich sequences and the peptide (Fig. S19).

We tested whether the DNA's higher affinity towards the peptide would lower the critical fuel concentration and increase the critical salt concentration (Fig. 6c, d, Supplementary Table 6). We found no difference in the critical fuel concentration required to form droplets between $G_8N_{14}G_8$ and the no-DNA control (8.3 vs. 9.2 mM) and an increase for $G_{10}A_{20}$ to 17.5 mM, consistent with the observed marginal reduction in droplet lifetime. In contrast, $G_8N_{14}G_8$ doubled the tolerance for additional sodium chloride, while $G_{10}A_{20}$ increased it even further to over 1 M. Nevertheless, the addition of salt decreased most of the turbidity with less than 100 mM additional salt, suggesting that the remaining turbidity may stem from DNA-peptide aggregates that are hard to dissolve due to the higher affinity of arginine peptides for guanine[68]. The combination of an unchanged critical fuel concentration and the increased critical salt concentration required to dissolve the droplets suggests that the overall affinity of droplet components is not significantly enhanced. Despite relatively high partitioning (Fig. S9), the main droplet components, peptide and poly-U, do not change their affinity. Instead, the droplets that form under the given anhydride conditions appear to be kinetically trapped. We hypothesize this is due to restricted peptide diffusion within the dense network. If the deactivated peptide cannot diffuse freely out of these aggregates, droplet dissolution would be impeded.

Furthermore, FRAP measurements confirmed our hypothesis. The peptide diffusivity decreased in the presence of both DNA sequences (Fig. 6e, f, Supplementary Table 11), whereas the poly-U diffusivity remained essentially unchanged in the presence of $G_8N_{14}G_8$, but was drastically reduced in the presence of $G_{10}A_{20}$ (Fig. S20, Supplementary Table 11). We conclude that both sequences reduce peptide mobility. In contrast, only the sequence containing adenines ($G_{10}A_{20}$) additionally restricts poly-U dynamics, consistent with hybridization-induced rigidification.

Confocal microscopy was used to investigate the morphological changes of the droplets due to their genotypes. With $G_8N_{14}G_8$, we observed aggregates, incomplete fusion, and large network structures (Fig. 6g, h) consistent with previous reports of poly-arginine aggregation with guanine-rich RNA[63]. After ten minutes, vacuoles formed—common during fuel starvation when droplets are large[24,45]—but unlike the control, these droplets retained a persistent shell and resisted dissolution. We attribute the vacuolization to the peptide deactivation outpacing peptide influx due to slowed peptide diffusion in the presence of DNA, causing core collapse and kinetic trapping. $G_{10}A_{20}$ showed a mixed morphology between $A_{30}$ and $G_8N_{14}G_8$, i.e., the droplets did not fuse properly and formed pearl-like aggregate strings similar to those with $A_{30}$, while the complete dissolution is delayed relative to $A_{30}$ (Fig. S21).

These combined results show that specific G-rich DNA motifs can markedly modulate droplet behavior. Sequences with guanine-rich termini induced kinetic trapping and network formation, likely due to strong peptide–guanine interactions. In contrast, a hybrid sequence $G_{10}A_{20}$ did not show trapping but displayed features of both G- and A-rich effects, including delayed dissolution and impaired fusion.

These combined results demonstrate that DNA sequences, as a primitive genotype, influence the behavior of our fuel-dependent synthetic cells. Next, we aimed to demonstrate that these benefits impact the synthetic cells in periodic fueling-starvation experiments. We found that droplets with a G-rich DNA genotype formed kinetically trapped, semi-fused shells. Upon refueling, we observed the regrowth of the semi-fused droplet network at the beginning of the cycle. This morphology differs from that observed upon the first fuel addition and from the refueling of regular droplets without DNA (Figs. 6h, S22). We conclude that our primitive genotype can influence the droplets over multiple rounds of fueling and starvation.

## Discussion

We investigated whether short DNA oligomers can function as rudimentary genotypes within fuel-dependent peptide- and RNA-based coacervate droplets. We screened random libraries, biased libraries,

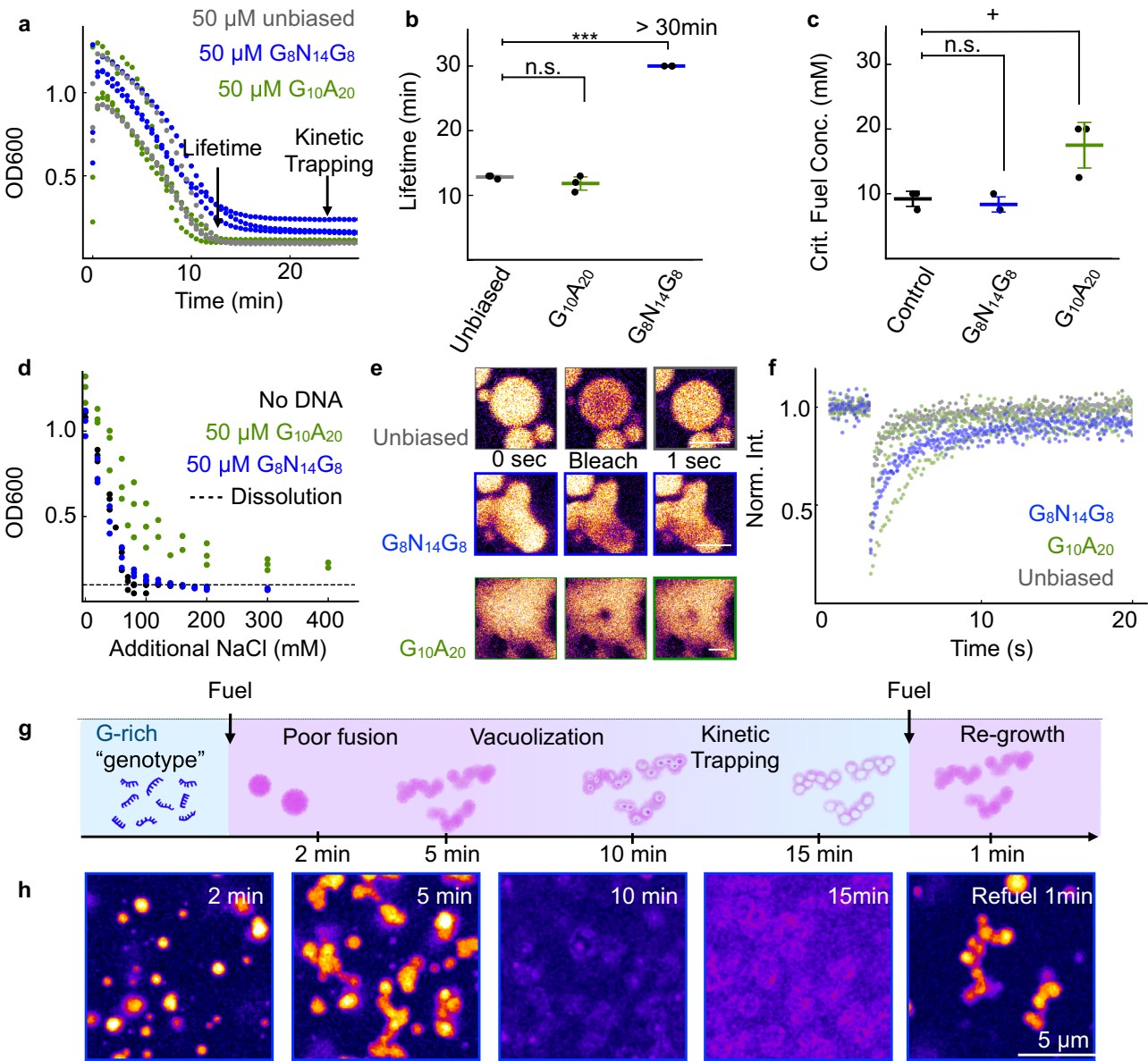

**Fig. 6 | Kinetic trapping of coacervate droplets is facilitated by guanine stretches within DNA sequences. a** Optical density at 600 nm (OD600) as a function of time for standard conditions in the presence of 50 μM DNA sequences containing guanine stretches (8 guanines on both sides: blue, 10 guanines on one side: green). **b** The lifetimes in the presence of $G_8N_{14}G_8$ (blue) and $G_{10}A_{20}$ (green) compared to the unbiased DNA (gray), as derived from the turbidity traces measured on the UV spectrometer. **c** The critical fuel concentration in the presence of G-rich DNAs ($G_8N_{14}G_8$:blue; $G_{10}A_{20}$:green) was recorded as the onset of turbidity on the UV-VIS spectrometer during titration with fuel and compared to a no DNA control sample (black). **d** The critical salt concentration in the presence of G-rich DNAs derived from the turbidity on the UV Vis spectrometer upon addition of salt to static droplets. ($G_8N_{14}G_8$: blue; $G_{10}A_{20}$: green; no DNA: black) **e**, **f** Confocal micrographs and

FRAP recovery traces of the NBD- tagged precursor peptide in the presence of G-rich DNA ($G_8N_{14}G_8$:blue; $G_{10}A_{20}$:green) or an unbiased library (gray) measured on the confocal microscope. Scalebars represent 10 μm. **g** Schematic representation of G-rich DNA kinetically trapping the assemblies and changing the morphology upon refueling. **h** Confocal micrographs imaged with 500 nM Sulforhodamine in the presence of $G_8N_{14}G_8$ at select timepoints in a fuel cycle and upon a second fuel addition. Scalebars represent 5 μm. All error bars show the standard deviation from the average (*N* = 3). *N* represents one biological replicate of one analyzed turbidity trace. Significance was determined via a two-sided Welch's *t*-test with the unbiased DNA sample as the control group (**b**) or the no DNA samples as the control group (**c**). *P*-values can be found in Supplementary Tables 5 and 6. Source data are provided as a Source Data file.

and specific DNA sequences for their ability to partition in the droplets, which is crucial for acting as a genotype. We found through sequencing that A-rich and G-rich sequences, especially when these bases were consecutive, partitioned well in the droplets. We tested how these well-partitioning sequences affected key phenotypic traits, including longevity, morphology, fusion, critical fuel needed to form droplets, and the amount of salt needed to dissolve the droplets. On the one hand, A-rich sequences weaken the interactions between RNA and peptide, which was adverse for the droplet's phenotype—they

lived shorter, required more fuel to form, and were less salt resistant. Despite their negative impact on the droplet, we were able to identify these sequences in the droplets, likely because we isolated the droplets early in the cycle. On the other hand, G-rich sequences, particularly those with guanine stretches at their termini, prolonged the droplet's longevity, even to a point where they did not dissolve. These droplets resisted dissolution despite the absence of fuel, pointing to strong interactions that hindered peptide diffusion. Interestingly, sequences combining G and A motifs exhibited mixed phenotypes, balancing

trapping and hybridization. Taken together, our results demonstrate the first steps toward coupling genotype and phenotype in fuel-driven synthetic cells. We believe that hybridization, rigidification, and diffusion inhibition, as interaction modes, are also transferable to other systems of active droplets.

We recognize that the DNA sequences in this study do not replicate and, therefore, cannot be considered true genotypes in a biological sense. Nevertheless, our results show that if such sequences could replicate, they could exert a strong influence on droplet behavior. In the context of periodic fueling–starvation experiments, droplets with "favorable" sequences—those that prolong lifetime or increase resilience to low fuel or high salt—would be more likely to survive a starvation period and resume growth when fuel becomes available again, as demonstrated in the refueling experiments. If these droplets were also capable of division, as we have recently demonstrated in related systems, these favorable sequences could be inherited by daughter droplets, allowing the persistence and spread of advantageous traits[12]. This scenario points toward a future in which integrating replicating genetic elements into fuel-dependent synthetic cells could enable true Darwinian evolution, with selection acting on genotype-defined phenotypes. To achieve that scenario, however, we would need our DNA sequences to replicate, which we envision is possible with rolling circle amplification or lesion-induced DNA amplification[38,69].

In summary, we demonstrate that short, non-replicating DNA oligomers can partition into fuel-driven, peptide–RNA coacervate droplets and modulate key phenotypic traits such as longevity, morphology, fusion behavior, and tolerance to fuel or salt stress. Systematic screening and sequencing revealed that consecutive adenine or guanine stretches—especially at sequence termini—drive strong partitioning, with A-rich sequences weakening peptide–RNA interactions and shortening lifetimes, while G-rich sequences induce kinetic trapping and markedly extend persistence. Our findings establish design rules for coupling sequence identity to droplet phenotype. We envision that, once combined with self-replicating DNA, these principles could enable droplets to survive starvation, endure environmental stress, and pass advantageous traits to progeny upon division—laying the groundwork for synthetic cells capable of Darwinian evolution.

## Methods
### Materials
We purchased 1-ethyl-3-(3-dimethylaminopropyl) carbodiimide (EDC), polyuridylic acid, potassium salt (poly-U), 2-(N-morpholino)ethanesulfonic acid (MES) buffer, sodium chloride, polystyrene sulfonate (17 kDa), triethylammonium acetate buffer, different amino acids, and Sulforhodamine from Sigma-Aldrich and used them without any further purification. General solvents were purchased from Sigma Aldrich in analytical or synthesis grade and used without further purification. Cy5-A$_{15}$ was purchased from biomers.net GmbH. Peptides were either ordered from Caslo or Biocat as the lyophilized TFA salt with a purity of >98% or synthesized on a peptide synthesizer according to the method described below. Self-synthesized peptides were characterized before use with electrospray ionization mass spectrometry (ESI-MS) in positive mode, and all peptides were characterized and their concentration verified by analytical HPLC as described in Supplementary Table 8. For library preparation, we used PEG-8000 from Carl Roth, RNase Cocktail from Invitrogen, and T4 DNA ligase, T4 pnk, NEBNext Ultra II Q5 Master Mix, NEBNext Multiplex Oligos for Illumina (Index Primer Set 3), and Monarch Spin PCR & DNA cleanup kit from New England Biolabs, NucleoMag NGS Clean-up and Size Selection beads from Macherey-Nagel, and PhiX Control v3 from Illumina. All DNA oligomers used in this study were ordered from IDT or Sigma Aldrich with standard desalting. Sequences are found in Supplementary Table 10 in the Supplementary Information.

**Peptide synthesis and purification.** Ac-F(RG)$_3$D-OH (Ac-Phe-Arg-Gly-Arg-Gly-Arg-Gly-Asp-OH) was synthesized on a 0.5 mmol scale with a Liberty Blue® peptide synthesizer from CEM by standard Fmoc-SPPS. Pre-loaded Wang resin (Fmoc-Asp(OtBu) loaded, 100–200 mesh, 0.67 mmol/g) was used. A double Fmoc-deprotection and double coupling cycle with microwave heating was applied. Therefore, Fmoc-removal was performed twice per amino acid (2 × 5 mL piperidine 20 v/v% solution in DMF, 90 °C, 90 s) before each amino acid was coupled twice (2 × 5 mL of 0.2 M amino acid in DMF, 2 × 2 mL 0.5 M DIC in DMF, 2 × 1 mL 1 M Oxyma pure in DMF, 90 °C, 120 s each cycle). Arginine couplings were performed for 240 s at 90 °C with the same reagent concentrations. Phenylalanine was used pre-acetylated. Sidechain deprotection and final cleavage from the resin were achieved with a solution of 2.5% water, 2.5% triisopropylsilan, and 95% trifluoroacetic acid for two hours under continuous agitation at room temperature. Solvents were evaporated under a constant nitrogen flow, and the residue was dissolved in a mixture of water and acetonitrile. Purification was done by preparative reversed-phase HPLC on an Agilent 1260 Infinity II setup (Agilent InfinityLab ZORBAX SB-C18 column (250 mm×21.2 mm, 5 μm particle size). Separation was performed on a linear gradient from 2 % to 98% acetonitrile in Mili Q water, both with 0.1% TFA at a flow rate of 20 ml/min. The fluorescently labeled peptide with NBD was prepared by reacting the peptide on the resin with two equivalents of NBD-Cl and 1.2 equivalents of DIPEA in 4 ml DMF overnight under constant agitation. Cleavage and purification was performed as described above, but with 3 hours of shaking at room temperature during the cleavage step[70].

**Sample preparation.** We prepared stock solutions of the peptide and peptide model by dissolving the peptides in MQ water, after which we adjusted the pH to pH 5.3. Stock solutions of EDC were prepared by dissolving the EDC powder in MQ water. We prepared the stock solutions of 0.5 M EDC freshly. Reaction cycles were started by adding the highly concentrated EDC to the acid solution. We carried out all experiments at 25 °C. Standard active droplet conditions were 23 mM peptide, 4.1 mM poly-U (monomer conc.), 30 mM EDC, 200 mM MES buffer (pH 5.3), and 50 μM DNA (strand conc.). Control samples without DNA were prepared with 5.6 mM poly-U (monomer conc.) to keep the overall charge constant. Standard static droplet conditions were 20 mM peptide, 3 mM peptide model, 4.1 mM poly-U, and 200 mM MES buffer (pH 5.3), and 50 μM DNA (strand conc.). Control samples without DNA were prepared with 5.6 mM poly-U (monomer conc.).

**DNA partitioning.** We determined the partitioning of different DNA sequences by measuring the DNA concentration in the supernatant after spinning down the droplets using analytical HPLC (HPLC Thermo Vanquish, Thermo DNAPac RP column particle size 4 μm 2.1 ×100 mm) with a gradient of acetonitrile in 100 mM triethylammonium acetate buffer and a flow rate of 0.3 ml/ min at 60 °C column temperature. The gradient ran from 6.25% acetonitrile to 25 % acetonitrile in 8 minutes, with a pre-equilibration of 5 minutes.

A 20 μL static droplet sample was prepared and spun down for 10 minutes. 18 μL supernatant was removed from the sample and transferred into a screw cap HPLC vial with a plastic inlet and incubated for at least 30 minutes in the HPLC. Samples of the solutions were injected without further dilution (injection volume: 1 μL) and tracked with a UV/Vis detector at 260 nm, and for the Cy3-A$_{30}$ additionally at 550 nm. For analysis the Chromeleon HPLC analysis software (Version 7) was used. Calibrations were made for each tested DNA library at three different concentrations. Each spin-down experiment for the libraries was performed in triplicate, apart from the additional A$_{30}$ and G$_8$N$_{14}$G$_8$. DNA partitioning coefficients were then calculated using the droplet volume determined with confocal microscopy, as described below. The final error was calculated from the standard deviation of

the measured DNA supernatant concentrations and the standard deviation of the droplet volume determination using error propagation.

**Sequencing Library preparation.** Detailed information can be found in the accompanying protocol paper.

**Sequencing Data analysis.** A detailed method for this can be found in the corresponding protocol paper. All scripts used to generate the final data and figures from the raw data can be found at https://github.com/hmutschler/Droplet-Sequencing-Analysis (https://doi.org/10.5281/zenodo.18597396).

We opted for paired-end sequencing, which means each molecule is read twice during sequencing, once from each end. Due to the short length of the library (including adapters >100 nt), this means that every position in the sequence is read twice and greatly reduces error frequency. After adapter trimming and length filtering, the R2 reads were discarded, as these reads are not present in the droplets (R1 represents all oligos from the $N_{30}$ library, while R2 is the reverse complement), and R2 reads were not further analyzed.

PolyG tails due to two color sequencing chemistry were accounted for in data processing, where we first trimmed adapters, usually deleting everything downstream of the $2^{nd}$ adaptor in each read, as well as when filtering for reads of exactly 30 nt length.

For the motifs, we decided to use an e-value of 0.05 as the cut-off for significance. The e-value corresponds to the p-value of these motifs appearing enriched at random, divided by the total number of motifs found by the STREME software.

**ITC measurements.** ITC titrations were performed on a MicroCal PEAQ-ITC instrument from Malvern Panalytical. All experiments were performed at 25 °C and with a control titration of the respective peptide concentration in 200 mM MES buffer (pH 5.3) into the same buffer without any DNA. All samples (cell and titration solutions) were prepared with the same concentration of MES buffer (200 mM at pH 5.3) to avoid additional dilution effects. The data was analyzed using a nonlinear least squares algorithm provided by the PEAQ-ITC analysis software. The component concentrations were chosen in a regime where no coacervation could be observed.

**UV/Vis Spectroscopy/OD measurements.** The UV/Vis measurements were carried out using a Multiskan GO (Thermo Fisher) microplate reader, which measures top-down absorption. Samples (either 20 μL or 50 μL) were directly prepared into a 384 or 96-well half-area plate (flat bottom) at 25 °C. Each experiment was performed for 30 minutes with turbidity measurements at 600 nm every 30 s and in triplicate. Lifetimes were determined at a threshold of 0.12; we define the droplet longevity at a threshold of 0.2.

**Critical Salt concentration determination.** We determined the critical salt concentration by titrating a 20 μL sample of static droplets with standard static droplet conditions with 0.5, 1, or 5 M sodium chloride solution in 10 or 20 mM steps. Volumetric dilution was neglected due to being <25% at the maximum, and the same for all samples. After each addition, the sample was mixed by pipetting up and down and left to incubate for 30 seconds until air bubbles had disappeared. Afterwards, the turbidity was measured at 25 °C. The titrations were performed in triplicate and each replicate was fit linearly. For the linear fit we use all absorption values > 0.12 to simplify fitting. We determined the critical fuel concentration as the average of the three concentrations where the extrapolated fits cross the threshold of 0.1. When dissolution was not linear and a linear fit was not possible due to aggregate formation (Fig. 6), we determine the critical salt concentration as the average of the first concentration with a measured absorption value < 0.1.

**Critical Fuel concentration determination.** We determined the critical fuel concentration by titrating a 20 μL sample with standard active droplet conditions with a 500 mM EDC solution in 2.5 (<12.5 mM EDC) and 5 mM steps (> 12.5 mM EDC). We determined the critical fuel concentration as the average of the first concentration where the turbidity crosses above the threshold of 0.1.

**Confocal Fluorescence Microscopy.** We used a Leica SP8 confocal microscope with a 63x oil immersion objective to image the droplets at different times. We prepared samples as described above, but with 500 nM Sulforhodamine B or the fluorescently tagged peptide as dye. 30 μL of the sample was deposited on a PVA-coated IBIDI μ-slide Angiogenesis Glass bottom slide. Samples were excited with a 552 nm laser and imaged at 565-635 nm with a HyD detector. Samples with the fluorescent peptide were excited with a 488 nm laser and imaged at 499-569 nm. The pinhole was set to 1 Airy unit. Images were taken either 20 μm above the bottom or on the glass. Micrograph images were analyzed with FIJI (Image J) and contrast and brightness adjusted for better visibility, for Fig. 6 a median filter was used for better visibility.

**Droplet volume determination.** The total droplet volume was determined from Z-projections of static droplet samples prepared in water in oil microreactors. These microreactors were screened as a z-stack after the droplets had mostly merged. The Z-projections of these images were analyzed with FIJI (ImageJ). From the area of the individual coacervates their volume was determined and added for the total droplet volume. The total sample volume was determined by measuring the maximum diameter of the water in oil droplet. The ratio of these values is given in percent. A total of 10 microreactors were screened and the average and standard deviation taken from that. The samples were prepared as static droplets using the standard conditions for static droplets (20 mM peptide, 3 mM peptide model, 4.1 mM poly-U, 200 mM MES (pH 5.3).

**Statistics and Reproducibility.** Measurements were taken from at least three distinct samples. For representative confocal images, experiments were performed at least three times. For all experiments where significance is indicated, we have performed a two-sided Welch's t-test for independent samples using SciPy in a Python (Version 3.13) script. A p-value below 0.05 is considered significant (*), a p-value of 0.1 is a slight trend (+). The degrees of freedom are calculated with the Welch-Satterthwaite equation. Tables with the exact p-values, t values, and calculated degrees of freedom can be found in the SI. Information on the significance of the cutoff during motif discovery can be found in the Sequencing Data analysis.

**Data collection and analysis.** To collect our data on the confocal, we used the Leica LAS X software Version 4.6, on the UV spectrometer, we used the SkanIt Software Version 6.0, on the HPLC, we used Chromeleon Version 7.3.2, and on the ITC, we used MicroCal PEAQ-ITC Control software(2022). The numerical generator can be found on GitHub at https://doi.org/10.5281/zenodo.18597396. For data analysis, we have used FIJI (ImageJ), Python (Version 3.13), Chromeleon Version 7.3.2, and Excel. The FRAP analysis script can be found on GitHub at https://doi.org/10.5281/zenodo.18597396.

**Fluorescence recovery after photobleaching (FRAP).** The diffusivity of droplet components inside the coacervates was calculated from fluorescence recovery after photobleaching experiments with the above-mentioned confocal in FRAP mode. We prepared static droplets with the standard conditions for static droplets mentioned above and imaged droplets close to the glass, waiting at least 5 min for the droplets to reach an adequate size. For the peptide diffusivity, we excited the fluorescent version of the precursor peptide, NBD-G(RG)$_3$D-OH, at

488 nm and detected it from 499 nm-569 nm with a PMT detector for at least 70 s. For the poly-U diffusivity, we annealed a short fluorescently tagged RNA oligomer, Cy5-$A_{15}$, to the poly-U and bleached at 638 nm, detecting from 650 nm-750 nm with a PMT detector for at least 180 s. Images were acquired at a resolution of 256 × 256 pixels, 700x scan speed (bidirectional scan), and between 8 and 10x zoom. The time intervals for the peptide bleach were 0.1342 s and 0.1902 s for the poly-U bleach. The bleach radius was chosen at 2.5 μm for the peptide and 1.0 μm for the poly-U and corrected for the calculation based on the post-bleach frame. Recovery data was double normalized using a FIJI macro script with the following equation:

$$F(t) = \frac{(I_t - B_t)}{(N_t - B_t)} \frac{(N_0 - B_0)}{(I_0 - B_0)} \tag{1}$$

F(t) represents the normalized fluorescence intensity at time point t. $I_t$ is the average intensity of the bleached ROI at a specific time point, $N_t$ is the average intensity of an unbleached ROI in a neighboring coacervate, and $B_t$ represents the average intensity of a ROI without any coacervates. $N_0$, $I_0$, and $B_0$ are the averaged pre-bleach intensities of the unbleached ROI, bleached ROI, and the background until the bleach frame. We fitted the data with the following exponential equation:

$$F(t) = a\left(1 - e^{(-bt)}\right) + c \tag{2}$$

With a representing the mobile fraction or the fluorescence at full recovery, b being related to the half recovery time, and c being the minimum intensity upon bleaching.

$$t_{1/2} = \frac{\ln(2)}{b} \tag{3}$$

The diffusion coefficient is calculated via the following calculation according to Soumpasis[71]:

$$D = \frac{0.224\, r^2}{t_{1/2}} \tag{4}$$

We have used the radius r from the postbleach frame; 0.224 is a numerically determined coefficient. We have determined the diffusion coefficient for each condition in triplicate.

## Reporting summary

Further information on research design is available in the Nature Portfolio Reporting Summary linked to this article.

## Data availability

The raw sequencing data generated in this study have been deposited in the Sequence Read Archive (SRA) of the National Center for Biotechnology Information (NCBI): https://www.ncbi.nlm.nih.gov/sra/PRJNA1424544. The other experimental data generated in this study are provided in the Supplementary Information/Source Data file. Source images are available upon request to Job Boekhoven due to their size; permanent access can be given with a one-month response time. Source data are provided with this paper.

## Code availability

All scripts used to generate the final data and figures from the raw sequencing data, as well as the numerical generator and the FRAP analysis script, can be found on GitHub at https://doi.org/10.5281/zenodo.18597396.

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

## Acknowledgements

We thank Monika Wenisch for providing the image analysis Macro used to analyze the FRAP data. The Mutschler lab is grateful for support from the European Research Council through the ERC Synergy Grant 101166888 (H.M.). The BoekhovenLab is grateful for support from the TUM Innovation Network – RISE, funded through the Excellence Strategy (J.B.). This research was conducted within the Max Planck School Matter to Life, supported by the German Federal Ministry of Education and Research (BMBF) in collaboration with the Max Planck Society (J.B.). Funded by the Deutsche Forschungsgemeinschaft (DFG, German Research Foundation) under Germany's Excellence Strategy - EXC-2094 – 390783311(J.B.). The BoekhovenLab is grateful for support from the European Research Council through ERC Starting Grant 852187 and ERC Consolidator Grant 101124380 (J.B.).

## Author contributions

C.M. and A.-L.H. both did experimental work, data analysis, conceptual planning, and writing, contributed equally, and have the right to list their names first in their CV or any bibliographic list. J.B. and H.M. supervised the research. The manuscript was written through the contributions of all authors. All authors have approved the final version of the manuscript.

## Funding

## Competing interests

The authors declare no competing interests.
