## [Transparent Peer Review file · Nature Communications]

DNA affects the phenotype of fuel-dependent coacervate droplets

Corresponding Author: Professor Job Boekhoven

Version 0:

Reviewer comments:

Reviewer #1

(Remarks to the Author)

In this manuscript, Machatzke & Holtmannspötter hypothesize that guest DNA strands in peptide/RNA droplets acts as a rudimentary genotype, dictating droplet properties (=phenotype). Peptide-RNA droplets are sustained by the condensation/hydrolysis cycle already well established in their research group. They vary the sequence of the guest DNA and develop an experimental setup in order to work with libraries of DNA, and to quantify partitioning for each individual sequence. The authors find that A-rich DNA libraries destabilize the droplets, also leading to quicker decay after activation. In turn, G-rich libraries do not change stability in comparison to an unbiased library.

The effect of A- and G- bases is explained based on partitioning and sequencing experiments and secondary structure predictions, showing that A- and G-rich motifs, and lack of secondary structure, both lead to higher partitioning. A-rich strands displace polyU from peptide complexes, while G-rich sequences introduce strong interactions with arginine residues. Besides longer decay times, this leads to higher stability to salt and towards coalescence, thereby demonstrating a coupling between genotype and phenotype in protocells.

I find the central question to be very interesting to the synthetic cell community, and the experimental approach very robust. Necessary controls are made and thermodynamic predictions are included for DNA, which is not common in this field. It introduces a new method allowing to work with DNA libraries, valuable for the protocell community. I also appreciate the good practice in indicating the protocol manuscript that will be submitted separately and in the author contributions statement.

I have some concerns regarding the scope of the discussions. I found it somewhat disappointing that in the Discussion, a long paragraph is dedicated to the limitations of the manuscript, instead of deepening the interpretation of results, namely:

- Whether the results can only be interpreted for this particular peptide/polyU mixture. Have you tested at least one other coacervate composition? It would contribute to your hypothesis that DNA-polyU interactions affect droplet properties. Conversely, have you tested RNA guests too (that could then be replicated)?
- Whether the "rudimentary genotype" can really be considered a guest in these coacervates, given that the concentration of DNA libraries is only 4x lower than that of polyU as far as I understood. It is also not clear if droplets containing DNA libraries all fall under the "liquid condensate" category?
- The schemes in figures 5G and A are too simplistic, and the manuscript would benefit from more discussion about the interactions that are being proposed - referencing to computational works in the literature for example.
- I am also missing reference to the following works:
 - <https://doi.org/10.1038/s41467-021-24111-x>, which discusses the relationship between droplet composition and phenotype (but growth rate);
 - <https://doi.org/10.1038/s41467-022-30158-1>, which introduces phenotype-genotype coupling and shows a perhaps comparable method for droplet sequencing, although focused on RNA;
 - <https://doi.org/10.26434/chemrxiv-2024-l40ch-v2>, which also investigates the effect of DNA/RNA sequence on droplet properties.
- You mention the need for replication for the analogy to a genotype truly hold. While I understand this is a challenge for another project, I find that the fuel-dependent aspect of the droplets here is underused and I don't understand why it is important for this work that the droplets are out of equilibrium. The results with the passive controls (amidated peptide) are already sufficient to show impact on stability. It would strengthen the manuscript to make use of that fuel dependency, for example by performing a second cycle of activation/decay.

Some minor comments:

- Some cross references are not correct, for example the call for Fig 1C on page 4

- The genotype/phenotype analogy is very interesting and appropriate, however, in some parts it seems the authors combine technical names with analogies, for example:
 - "We tested how these well-partitioning sequences affected key phenotypic traits, including longevity, morphology, fusion, critical fuel needed to form droplets" (longevity may be too much personification)
- The introduction is very long, and with several references cited in a block, for aspects that I don't find so relevant to the present work. For example, there are six references for the concept of fuel-dependent synthetic cells, all regarding the same system, and ultimately, fuel dependency is not the main result on the present manuscript. I would ask the authors to either select, or specify better why some references are cited, especially because so many are from the same group.
- I'm a bit confused that the plots in Fig2 C and D are line plots and not a dispersion - is this a fit, or simulation?

(Remarks on code availability)

Reviewer #2

(Remarks to the Author)
Manuscript NComms-25-69143:

This manuscript by Machatzke et al. show the effect of DNA on RNA-peptide coacervates. In this study, the authors, study the effect of 14 different short DNA sequences and the number and location of Guanines and Adenines on the partitioning and FRAP recovery of the droplet. The coacervate system (RNA-peptide) is based on active peptides that the Boekhoven lab has been utilizing for the preparation of active droplets and in this work the authors contextualise their study with regard to genotype-phenotype coupling with future studies focused on the effect of activity on DNA propagation. This work is actually a study on the incorporation of different DNA into coacervate droplets. Without studies which connect active droplets behaviour with DNA propagation this work falls short of the general suitability for Nature communications and would be better suited to a more specialized journal. Furthermore, the authors should ensure that the arguments and interpretation are fully and correctly supported by experimental data.

Please find below for more detailed comments:

1. The Schematic in figure is misleading given the presented data. Where do they see growth and decay of the droplets? One needs to take care about the interpretation of the OD data. A decrease in turbidity could also be associated with the droplets coming out of the laser line due to gravity and not from a smaller size of compartment. Further the interpretation of the maximum turbidity should also be reconsidered as there is not a significant difference in these samples. In light of this the discussion regarding longevity should be re-evaluated. What exactly do they mean with longevity given the experimental data? Please include the methods for the turbidity measurements.
2. Provide some reasoning as to the increase in partitioning between G and A rich DNA.
3. It was interesting to see that they observe a low general partitioning of the DNA into these droplets. Can a rationale be provided for this?
4. The authors make a statement: "Based on the thermodynamic secondary structure prediction, we conclude that intramolecular structures in sequences are detrimental to their partitioning into the droplets in line with previous work.⁵⁴" References to other work also needs to be included here, both with DNA and RNA
5. Is there a difference in DNA partitioning of active peptide droplets compared to non-active droplets? This can also give some insight into specific peptide and DNA interactions.
6. Figure 2: authors should specify that this is the coacervate phase vs supernatant as opposed to droplets vs supernatant. How significant are these findings given that there are no error bars?
7. To provide more evidence for their statements regarding A30 and polyU interactions- can they directly show DNA-PolyU hybridization?
8. Does the peptide leave the coacervate upon addition of DNA?
9. Can the authors provide more evidence that the droplets do not want to coalesce. For example, the authors can undertake optical tweezer experiments. Could the authors rule out effects from surfaces that can lead to droplet dissolution? Rationale for these DNA sequences in terms of the community should be discussed.

(Remarks on code availability)

Reviewer #3

(Remarks to the Author)

In this manuscript, Machatzke, Holtmannspotter et al present work demonstrating that short DNA sequences (and their contents) can drive the behavior of synthetic cells. Given my expertise in next generation sequencing, my review is specific to the methods and conclusion derived from the NGS experiments.

With respect to the NGS methods, my concern is that the authors do not provide evidence that their novel sequencing method has bias. Specifically, during adapter ligation, the authors use the SRSly splint adapters, which contain a single-stranded overhang composed of random nucleotides to bind to their ssDNA oligos. These splint adapters, although they contain random bases, preferentially bind to certain ssDNA motifs. PCR can also introduce GC-biases. The authors should demonstrate the extent of the bias of their library preparation and that this bias does not alter the conclusions of the manuscript. To clarify, all libraries preparation methods and sequencing experiments have some form of bias, but the degree of bias can be acceptable if the signal of interest is stronger than the bias.

In addition, fastQ files can contain a lot of junk reads that are filtered by DNA aligners during mapping. Given that there is no genome alignment (understandably), how did the authors mitigate this issue? For example, fastQ files from two-color chemistry sequencers (for eg iSeq) can contain poly-Gs that are systematic of the sequencer and not of the library. See here: https://support.illumina.com/content/dam/illumina-support/help/Illumina_DRAGEN_Bio_IT_Platform_v3_7_100000141465/Content/SW/Informatics/Dragen/PolyG_Trimming_FDG.htm

How did the authors mitigate this issue? Was it addressed?

Of note: it's not clear to me why R2 is discarded in the bioinformatic process. I agree with the authors that R1 represents the sequence of the original droplet, but R2 is the reverse complement and could be used as well. Why do paired-end sequencing just to discard R2?

While my opinion on the novelty and general interest should be taken with a grain of salt given that I am not an expert in the field of synthetic cells, I would like to mention that I found this manuscript very enjoyable and informative.

(Remarks on code availability)

While I did not test the code, I read through the GitHub page as well as the additional manuscript they authors provided on the molecular and computational sequencing methods. I found the documents to be extremely well documented. The bioinformatic analysis pertaining to the sequencing experiments seem adequate (adapter trimming using cutadapt, fastQC to assess sequencing quality, seqkit for fastQ manipulation).

Version 1:

Reviewer comments:

Reviewer #1

(Remarks to the Author)

The authors have addressed most of the points I found unclear during revision. They include now a control with a non-RNA based coacervate (peptide/PSS) that strengthens the argument for the role of hybridization. They have performed a cycle of refueling for the G-rich system (showing that growth resumes indeed), which justifies the use of an activation/decay network. They have also worked on Figures 2 and 5 and addressed the block/self-citations mentioned. The discussion of the work in light of the literature still could be longer and deeper in my opinion; and like for reviewer 2, the term longevity can be distracting, but those are not reasons to revise the manuscript. I consider the manuscript suitable for publication.

(Remarks on code availability)

Reviewer #2

(Remarks to the Author)

Review of "DNA affects the phenotype of fuel-dependent coacervate droplets".

The work focuses on the addition of DNA sequences on active peptide-RNA coacervate droplets and studies the effects of adenine and guanine rich motifs on the properties of the droplets. The authors have added additional experiments, especially important are the experiments where they include DNA into a refueled system. The comparative results with and without DNA can be entirely included into the main text.

Given other studies which have looked at nucleic acid uptake and the change in phenotype in coacervates, this study looks at the same problem but from a different coacervate system and with a focus on adenine and guanine. Therefore the conceptual and scientific novelty of this work should be questioned but I can leave that to the editor to decide whether this work is suitable for the journal.

In direct response to the rebuttal, please see the comments to the following points.

1. Further, the interpretation of the maximum turbidity should also be reconsidered as there is not a significant difference in these samples.

The difference in maximum turbidity has been significant, as indicated in the Figure.

Unless we are discussing different figures or different meanings from significance, I would disagree with this point. In figure 1C the turbidity differences are not significant. The figure shows that in the absence of DNA, G and unbiased turbidities' are all within error of each other. In addition, unbiased, T-rich, C-rich and A rich are within error.

Furthermore, the authors really need to take care in interpreting the data. For example, for Figure S10, The authors say "To determine the critical salt concentration, we used our in-equilibrium model droplets and monitored the amount of additional NaCl required to dissolve our droplets critical salt concentration was 80 mM NaCl, which dropped to 50 mM NaCl in the presence of 50 μ M A30."

Whilst the conclusion is correct the numbers are not. It is 65 \pm 2.5 mM and 45 \pm 2.5 mM respectively. All data and interpretation should be carefully checked.

6. Figure 2: authors should specify that this is the coacervate phase vs supernatant as opposed to droplets vs supernatant. How significant are these findings given that there are

no error bars?

We thank you for adding more precision to our wording in this case. We have experimented with duplicates and obtained very similar results, and present the dataset with the higher number and quality of reads, as is not uncommon in the literature. The significance of the results we show in Figure 2E comes from the statistical analysis of the dataset.

The Authors did not address this point about supernatant and droplet. Do they mean droplets or condensed phase?

(Remarks on code availability)

Reviewer #3

(Remarks to the Author)

The authors have adequately responded to my comments. The NGS experiments and subsequent bioinformatic analysis appear appropriate for this study. The code is well-documented and reproducible.

(Remarks on code availability)

I did not test the code but read through the documentation during my first review. The code is very well-documented and reproducible. I have no concerns about this section and commend the authors on their attention to detail.

Version 2:

Reviewer comments:

Reviewer #2

(Remarks to the Author)

The authors have addressed all the points.

(Remarks on code availability)

Open Access This Peer Review File is licensed under a Creative Commons Attribution 4.0 International License, which permits use, sharing, adaptation, distribution and reproduction in any medium or format, as long as you give appropriate credit to the original author(s) and the source, provide a link to the Creative Commons license, and indicate if changes were made. In cases where reviewers are anonymous, credit should be given to 'Anonymous Referee' and the source.

REVIEWER COMMENTS

Reviewer #1 (Remarks to the Author):

In this manuscript, Machatzke & Holtmannspötter hypothesize that guest DNA strands in peptide/RNA droplets acts as a rudimentary genotype, dictating droplet properties (=phenotype). Peptide-RNA droplets are sustained by the condensation/hydrolysis cycle already well established in their research group. They vary the sequence of the guest DNA and develop an experimental setup in order to work with libraries of DNA, and to quantify partitioning for each individual sequence. The authors find that A-rich DNA libraries destabilize the droplets, also leading to quicker decay after activation. In turn, G-rich libraries do not change stability in comparison to an unbiased library.

The effect of A- and G- bases is explained based on partitioning and sequencing experiments and secondary structure predictions, showing that A- and G-rich motifs, and lack of secondary structure, both lead to higher partitioning. A-rich strands displace polyU from peptide complexes, while G-rich sequences introduce strong interactions with arginine residues. Besides longer decay times, this leads to higher stability to salt and towards coalescence, thereby demonstrating a coupling between genotype and phenotype in protocells.

I find the central question to be very interesting to the synthetic cell community, and the experimental approach very robust. Necessary controls are made and thermodynamic predictions are included for DNA, which is not common in this field. It introduces a new method allowing to work with DNA libraries, valuable for the protocell community. I also appreciate the good practice in indicating the protocol manuscript that will be submitted separately and in the author contributions statement.

Thank you for the time invested in our work.

I have some concerns regarding the scope of the discussions. I found it somewhat disappointing that in the Discussion, a long paragraph is dedicated to the limitations of the manuscript, instead of deepening the interpretation of results, namely:

- Whether the results can only be interpreted for this particular peptide/polyU mixture. Have you tested at least one other coacervate composition? It would contribute to your hypothesis that DNA-polyU interactions affect droplet properties. Conversely, have you tested RNA guests too (that could then be replicated)?

Thank you for these ideas! We have now tested the addition of A₃₀ in a system using the same peptide and polystyrene sulfonate as a polyanion. This system has been extensively characterized in our group. The main difference is a higher affinity of the peptide for this artificial polyanion (K_d 3.2 vs 130 μM).

As expected, without the possibility of hybridizing to the droplet material, no change in the lifetime was observed upon addition of this DNA, strengthening the hypothesis of DNA-polyU hybridization.

- Whether the "rudimentary genotype" can really be considered a guest in these coacervates, given that the concentration of DNA libraries is only 4x lower than that of polyU as far as I understood. It is also not clear if droplets containing DNA libraries all fall under the "liquid condensate" category?

Our DNA partitioning shows a broad range, from 5% to 99% of the DNA inside the droplets, indicating that the transition from guest to part of the system is highly dependent on the added DNA. While we show in Figure 3B that even lower concentrations than our standard 50 μM can affect droplet properties, we deliberately chose high DNA concentrations for the

phenotype experiments to showcase the impact of the different interaction modes explicitly. We believe that even at lower concentrations, the interaction modes would be the same and even transferable to different systems.

You are right that not all our droplets are equally liquid—some fuse rapidly, others are more aggregated. We have highlighted that clearly in the main text twice.

- The schemes in figures 5G and A are too simplistic, and the manuscript would benefit from more discussion about the interactions that are being proposed - referencing to computational works in the literature for example.

We include an additional sentence and cite additional foundational papers on the strength and prevalence of interactions between guanine and arginine, including simulations. Our specific case —single-stranded DNA with guanine repeats, combined with short arginine-containing peptides —is an uncommon target for such simulations. The ubiquity of the interactions between arginine and arginine-glycine repeats and guanine (-repeats) in nature also clearly indicates a preferential binding.

We have changed the focus of the scheme in Figure 5 to describe better the phenotypical changes we see as well as the impact of refueling.

- I am also missing reference to the following works:

- <https://doi.org/10.1038/s41467-021-24111-x>, which discusses the relationship between droplet composition and phenotype (but growth rate);
- <https://doi.org/10.1038/s41467-022-30158-1>, which introduces phenotype-genotype coupling and shows a perhaps comparable method for droplet sequencing, although focused on RNA;
- <https://doi.org/10.26434/chemrxiv-2024-l40ch-v2>, which also investigates the effect of DNA/RNA sequence on droplet properties.

Thank you for pointing us in the right direction. We added the works at the appropriate places and mentioned the comparable sequencing method directly in the text.

- You mention the need for replication for the analogy to a genotype truly hold. While I understand this is a challenge for another project, I find that the fuel-dependent aspect of the droplets here is underused and I don't understand why it is important for *this* work that the droplets are out of equilibrium. The results with the passive controls (amidated peptide) are already sufficient to show impact on stability. It would strengthen the manuscript to make use of that fuel dependency, for example by performing a second cycle of activation/decay.

You are absolutely right. What's the point in delaying dissolution if we don't refuel the experiments? A limitation of our system is that multiple cycles of activation and decay often lead to kinetic trapping due to the accumulation of waste products and metastability (See Holtmannspötter et al., 2024; "metastability" in Donau et al., 2020; Supporting Fig. 22).

We agree that the combination of longevity and refueling could be better exploited. Thus, we have performed additional experiments comparing the refueling behavior of kinetically trapped assemblies in the presence of G8N14G8 with that in our no-DNA control. We refuel the control droplets in the last minute of the cycle (13 min) and the kinetically trapped assemblies after 15 minutes, when only the shells remain. We see that in both cases, the surviving assemblies regenerate and form the next generation. Without DNA, we see droplets of different sizes that start fusing and become uniform over time. In the presence of G8N14G8, we observe that, only 1 minute after the second fuel addition, droplets are incorporated into the network structure of the previous generation, maintaining a distinct

network structure and thereby transporting information into the next generation. We added that data to the main text and the SI.

Some minor comments:

- Some cross references are not correct, for example the call for Fig 1C on page 4

Thank you for bringing this to our attention. We have changed the call to Fig. 1D.

- The genotype/phenotype analogy is very interesting and appropriate; however, in some parts it seems the authors combine technical names with analogies, for example:

○ "We tested how these well-partitioning sequences affected key phenotypic traits, including longevity, morphology, fusion, and critical fuel needed to form droplets" (longevity may be too much personification)

We believe that longevity is a term that shows what we want the reader to see, the other terms are standard descriptors in the field.

- The introduction is very long, and with several references cited in a block, for aspects that I don't find so relevant to the present work. For example, there are six references for the concept of fuel-dependent synthetic cells, all regarding the same system, and ultimately, fuel dependency is not the main result on the present manuscript. I would ask the authors to either select, or specify more clearly which references are cited, especially since so many are from the same group.

We have addressed the block citations and reduced them to the most relevant ones to help understand our system.

- I'm a bit confused that the plots in Fig2 C and D are line plots and not a dispersion - is this a fit, or simulation?

The lines between data points in Figure 2C just connect the datapoints—they guide the eye. Since every data point represents a single position in the ssDNA 30mer, no simulation or fitting can be done, because no data exists between the points. We chose this representation to make it easier to follow individual nucleotides across the entire sequence and to emphasize the significance of the termini for guanines.

In Figure 2D, we generated the minimum free energy for the entire dataset using the in silico folding algorithm RNAfold. This algorithm simulates the folding of sequences under specific conditions, as outlined in the SI or the accompanying method paper. The minimum free energy was then binned for plotting and re-plotted as a line graph connecting the individual bins.

We have changed Figure 2C to a line with additional individual data points, illustrating the discrete values at each position. We also changed Figure 2D to a bar graph, with each bin represented by a single bar.

Reviewer #2 (Remarks to the Author):

Manuscript NComms-25-69143:

This manuscript by Machatzke et al. show the effect of DNA on RNA-peptide coacervates. In this study, the authors, study the effect of 14 different short DNA sequences and the number and location of Guanines and Adenines on the partitioning and FRAP recovery of the

droplet. The coacervate system (RNA-peptide) is based on active peptides that the Boekhoven lab has been utilizing for the preparation of active droplets and in this work the authors contextualise their study with regard to genotype-phenotype coupling with future studies focused on the effect of activity on DNA propagation.

Thank you for the time invested in our work.

This work is actually a study on the incorporation of different DNA into coacervate droplets. Without studies which connect active droplets behaviour with DNA propagation this work falls short of the general suitability for Nature communications and would be better suited to a more specialized journal. Furthermore, the authors should ensure that the arguments and interpretation are fully and correctly supported by experimental data.

Please find below for more detailed comments:

1. The Schematic in figure is misleading given the presented data. Where do they see growth and decay of the droplets? One needs to take care about the interpretation of the OD data. A decrease in turbidity could also be associated with the droplets coming out of the laser line due to gravity and not from a smaller size of compartment.

We thank the reviewer for their attention to experimental detail. Regarding the interpretation of the OD data, we have added the detail that we measure the top-down absorption in a microplate. There should not be a decrease in turbidity due to droplets falling out of the laser beam, as can occur when measuring horizontally in a cuvette.

Moreover, to complement the OD data, we have now included two representative videos comparing the earlier droplet death in the presence of A30 with a control without DNA. We image the droplets on the glass of an ibidi chamber, avoiding them from falling out of plane and imaging until complete dissolution. As shown in the video, the droplets do indeed disappear earlier in the presence of A30. Without DNA, we observe growth by coalescence, followed by shrinking towards the end of the cycle.

In light of this the discussion regarding longevity should be re-evaluated. What exactly do they mean with longevity given the experimental data?

With longevity, we mean falling below an absorbance threshold, as indicated by the dotted line in the figure.

Further, the interpretation of the maximum turbidity should also be reconsidered as there is not a significant difference in these samples.

The difference in maximum turbidity has been significant, as indicated in the Figure.

Please include the methods for the turbidity measurements.

We have added the detail that we measure the top-down absorption in a microwell plate.

2. Provide some reasoning as to the increase in partitioning between G and A rich DNA.

Thank you for pointing this out. A requirement for basepairing and therefore high partitioning into the droplets is a minimum of seven consecutive adenines. In the A rich DNA, only about a third of sequences fulfill the criteria. We assume that the guanine-arginine interaction is less position-dependent and sequence-dependent, leading to higher partitioning as well as

being driven by a high peptide concentration. We have now further clarified this in the main text around Figure 1.

3. It was interesting to see that they observe a low general partitioning of the DNA into these droplets. Can a rationale be provided for this?

We observe a broad range of partitioning, from 5% to 99% of the DNA into the droplets. We believe that the DNA on the lower end of the partitioning, the random 30mer DNA, which has no specific interaction with the droplet material, is mainly driven by electrostatic driving forces. In that case, the DNA competes with the much longer poly U strands for a limited concentration of anhydride and is disadvantaged by multivalency effects. We have added this discussion to the main text.

4. The authors make a statement: “Based on the thermodynamic secondary structure prediction, we conclude that intramolecular structures in sequences are detrimental to their partitioning into these droplets in line with previous work.⁵⁴” References to other work also needs to be included here, both with DNA and RNA

We have added two citations that show the exclusion of folded and duplex DNA in this system. Since the uptake of folded sequences in different systems (particularly poly lysine RNA droplets) is more complicated than a simple exclusion based upon folding energy, we have amended the sentence to be limited to this arginine-based system.

5. Is there a difference in DNA partitioning of active peptide droplets compared to non-active droplets? This can also give some insight into specific peptide and DNA interactions.

We measured partitioning in non-active droplets, as it depends on the anhydride level. Electrostatics are the main driving force for general DNA partitioning and the concentration of positive charge changes over the course of the cycle due to the fuel being consumed and the anhydride level decreasing. It is therefore easier to measure reproducible partitioning in non-active droplets than at a specific time point in the cycle. To compare the partitioning of different DNAs, we decided to minimize this possible error source and used non-active droplets with a fixed model-anhydride concentration.

To show the effect different timepoints in the cycle have on partitioning we have included an additional figure on DNA recovery at different timepoints. We can see that a reduced anhydride level at seven minutes corresponds to lower partitioning than at minute one. We acknowledge that using the non-active droplets in this case is a simplification, and further investigations on the differences in the mechanisms of partitioning in active droplets, which contain more complicated fluxes, should be done. We believe this is beyond the scope of the manuscript at this point.

6. Figure 2: authors should specify that this is the coacervate phase vs supernatant as opposed to droplets vs supernatant. How significant are these findings given that there are no error bars?

We thank you for adding more precision to our wording in this case. We have experimented with duplicates and obtained very similar results, and present the dataset with the higher number and quality of reads, as is not uncommon in the literature. The significance of the results we show in Figure 2E comes from the statistical analysis of the dataset.

7. To provide more evidence for their statements regarding A30 and polyU interactions- can they directly show DNA-PolyU hybridization?

We have made this clearer in the text and provided additional evidence in the form of two additional SI figures. In these, we show the absence of lifetime changes in a non-hybridizing system (a different polyanion) and in a non-denaturing gel, where the hybridization of A30 can be clearly seen. The A30 is tagged with Cy-3 dye and run once without and once with 5x excess polyU. We observe that only the hybridizing A30 is stuck to the long polyU strands on top of the 14% acrylamide gel.

8. Does the peptide leave the coacervate upon the addition of DNA?

We do not consider this a general mechanism for every DNA. In the original experiments, DNA has been present from the beginning, and we assume that most interactions, especially hybridization, occur even before droplet formation, as shown by the non-denaturing gel. Additionally, in the case of hybridization of the added DNA with the RNA, we still observe preferred partitioning of the peptide in the coacervates at the 5-minute mark. This can be seen in the micrographs in Figure 4 as well as Figure S11-S14, which use the NBD tagged version of the peptide for staining. We do acknowledge that the polyanion's affinity towards the peptide is reduced in the case of hybridization. This is why the micrographs of the hybridizing DNAs appear less bright (i.e., less peptide within the assemblies). This phenomenon has been present from the beginning of the droplet cycle.

To complement our research on this and to investigate behavior when adding DNA later in the cycle, we have tested adding hybridizing A₃₀ DNA at different points after droplet formation and added figure in the SI. Adding the hybridizing DNA multiple minutes into the cycle still shortens the lifetime; however, this effect is not immediate upon DNA addition, as we would expect if the peptide truly left the coacervates. We therefore can't confirm this mechanism and rely on a gradual loss of affinity due to hybridization.

9. Can the authors provide more evidence that the droplets do not want to coalesce. For example, the authors can undertake optical tweezer experiments. Could the authors rule out effects from surfaces that can lead to droplet dissolution? Rational for these DNA sequences in terms of the community should be discussed.

Since our regular droplets show fusion (see movie supp movie 1), we assume this question refers to the formation of network-like structures in the presence of DNA containing G repeats, as well as the short-lived aggregates in the presence of A30. We have added a video of the droplet lifetime in the presence of A30 and without DNA, showing droplet fusion for the control and the absence of it for the A30 sample. For the suggested optical tweezer experiments, our kinetically trapped assemblies have an unsuitable arrested shell-like structure, while the transient A30-polyU aggregates are sadly too short-lived (5 minutes) to do the suggested analysis. Our FRAP data and the additional video provide sufficient proof of the absence of fusion. We chose the sequences whose phenotypical impact we examined in detail based on their prevalence in the sequencing data.

Reviewer #3 (Remarks to the Author):

In this manuscript, Machatzke, Holtmannspotter et al present work demonstrating that short DNA sequences (and their contents) can drive the behavior of synthetic cells. Given my expertise in next-generation sequencing, my review is specific to the methods and conclusions derived from the NGS experiments.

Thank you for taking the time to review this work. We greatly appreciate your input.

With respect to the NGS methods, my concern is that the authors do not provide evidence that their novel sequencing method has bias. Specifically, during adapter ligation, the authors use the SRSLY splint adapters, which contain a single-stranded overhang composed of random nucleotides to bind to their ssDNA oligos. These splint adapters, although they contain random bases, preferentially bind to certain ssDNA motifs. PCR can also introduce GC-biases. The authors should demonstrate the extent of the bias of their library preparation and that this bias does not alter the conclusions of the manuscript. To clarify, all libraries preparation methods and sequencing experiments have some form of bias, but the degree of bias can be acceptable if the signal of interest is stronger than the bias.

Thank you for raising this important point. Bias during library preparation was one of our key considerations when adapting this method. While no sequencing approach is entirely free of bias, we relied on the established SRSLY ligation protocol as a foundation. In addition, all sequencing data in this study are analyzed in a comparative framework: each sample is measured against a control processed with the same protocol. In this way, any systematic bias introduced during adaptor ligation or PCR (e.g., GC-content bias) would be present across all conditions and thus cancel out when comparing relative enrichment. Only sample-intrinsic differences remain after these comparisons, ensuring that the observed effects are not artifacts of the method. We have highlighted this comparative approach in the manuscript.

In addition, fastQ files can contain a lot of junk reads that are filtered by DNA aligners during mapping. Given that there is no genome alignment (understandably), how did the authors mitigate this issue? For example, fastQ files from two-color chemistry sequencers (for eg iSeq) can contain poly-Gs that are systematic of the sequencer and not of the library. See here: https://support.illumina.com/content/dam/illumina-support/help/Illumina_DRAGEN_Bio_IT_Platform_v3_7_1000000141465/Content/SW/Informatics/Dragen/PolyG_Trimming_fDG.htm

How did the authors mitigate this issue? Was it addressed?

Thank you for pointing us to this issue. While this specific sequencing artifact was not initially considered, our processing pipeline effectively eliminates it. All reads are first trimmed for adaptors, after which we retain only those that are exactly 30 nucleotides in length—the expected library size. Since our sequences are shorter than the total read length, poly-G tails from two-color sequencers only occur beyond the second adaptor and are therefore removed during trimming. Any residual mis-trimmed reads would also be excluded during the fixed-length filtering step. Together, these safeguards ensure that poly-G artifacts do not influence our results.

Of note: it's not clear to me why R2 is discarded in the bioinformatic process. I agree with the authors that R1 represents the sequence of the original droplet, but R2 is the reverse complement and could be used as well. Why do paired-end sequencing just to discard R2?

We appreciate the opportunity to clarify this point and updated the manuscript to explain this in more detail. We used paired-end sequencing primarily as a built-in proofreading mechanism for our short libraries: each molecule is read in full from both ends, and any discrepancy between R1 and R2 during trimming leads to exclusion by cutadapt. After this step, we focus only on R1 in downstream analyses because it directly corresponds to the original droplet sequence. Incorporating R2 would not add independent information about droplet partitioning and could dilute relevant signals. Thus, R2 is used for error control but not for sequence-based analysis.

While my opinion on the novelty and general interest should be taken with a grain of salt given that I am not an expert in the field of synthetic cells, I would like to mention that I found this manuscript very enjoyable and informative.

Thank you for the compliment and the constructive feedback.

Reviewer #3 (Remarks on code availability):

While I did not test the code, I read through the GitHub page as well as the additional manuscript they authors provided on the molecular and computational sequencing methods. I found the documents to be extremely well documented. The bioinformatic analysis pertaining to the sequencing experiments seem adequate (adapter trimming using cutadapt, fastQC to assess sequencing quality, seqkit for fastQ manipulation).

Thank you very much for the kind words. The manuscript for the method paper is currently under peer review. All sequencing data used for the method manuscript paper can be found here: <https://doi.org/10.17877/TUDODATA-2025-MFE1EMF9>. The code has been generalized to a broader audience and still produces the same results as presented in this work.

Reviewer 2

The work focuses on the addition of DNA sequences on active peptide-RNA coacervate droplets and studies the effects of adenine and guanine rich motifs on the properties of the droplets. The authors have added additional experiments, especially important are the experiments where they include DNA into a refuelled system. **The comparative results with and without DNA can be entirely included into the main text.**

Given other studies which have looked at nucleic acid uptake and the change in phenotype in coacervates, this study looks at the same problem but from a different coacervate system and with a focus on adenine and guanine. Therefore the conceptual and scientific novelty of this work should be questioned but I can leave that to the editor to decide whether this work is suitable for the journal. In direct response to the rebuttal, please see the comments to the following points.

1. **Further, the interpretation of the maximum turbidity should also be reconsidered as there is not a significant difference in these samples.**

The difference in maximum turbidity has been significant, as indicated in the Figure.

Unless we are discussing different figures or different meanings from significance, I would disagree with this point. In figure 1C the turbidity differences are not significant. The figure shows that in the absence of DNA, G and unbiased turbidities' are all within error of each other. In addition, unbiased, T-rich, c-rich and A rich are within error.

We have now added to the main text and the figure that the differences of no-DNA control, G-rich and unbiased libraries are indeed not significant. We have deleted one sentence that we agree wrongly suggested that, and further revised our phrasing to be more careful and better reflect the results. The other experiments significantly differ in maximum turbidity compared to the no-DNA control as determined by a two-sided Welch T-test. We have clarified that in the figure caption and explicitly linked to the statistical analysis in Table S1.

6. **Figure 2: authors should specify that this is the coacervate phase vs supernatant as opposed to droplets vs supernatant. How significant are these findings given that there are no error bars?**

We thank you for adding more precision to our wording in this case. We have experimented with duplicates and obtained very similar results, and present the dataset with the higher number and quality of reads, as is not uncommon in the literature. The significance of the results we show in Figure 2E stems from the statistical analysis of the dataset.

The Authors did not address this point about supernatant and droplet. Do they mean droplets or condensed phase?

The droplets have been combined into one condensed phase upon centrifugation, so we agree that the formulation "a droplet's genotype" is misleading in our case, as the individual droplets do no longer exist. We have before approached the terms condensed/coacervate phase and droplet phase as equivalent, but we see how this can be confusing. To be clearer, we have explicitly stated the spinning down methodology in the text and added more precise wording throughout the paragraph, now calling it the *combined* droplet phase.

Furthermore, the authors really need to take care in interpreting the data. For example, for Figure S10, The authors say “To determine the critical salt concentration, we used our in-equilibrium model droplets and monitored the amount of additional NaCl required to dissolve our droplets critical salt concentration was 80 mM NaCl, which dropped to 50 mM NaCl in the presence of 50 μ M A30. “

Whilst the conclusion is correct the numbers are not. It is 65 +/- 2.5 mM and 45 +/- 2.5 mM respectively. All data and interpretation should be carefully checked.

We thank the reviewer for bringing this to our attention. We have re-examined the interpretation of the Figure and, to be more accurate, have added linear fits to determine the exact point at which the threshold is crossed in this figure, as well as in Figure S2. We have updated the numbers in the text and the method section and checked every other interpretation and value mentioned. This has led us to update the average lifetime values shown in Figures 3, 4, and 5, as well as the marked values in the text. These are very minor changes that do not alter any conclusions or significance calculations.

We are grateful that the reviewer took the time to thoroughly review the complete manuscript and supporting information, which is what makes a manuscript stronger and more precise.